

**Size-resolved aerosol composition at an urban and a rural site in**
**the Po Valley in summertime: implications for secondary aerosol**
**formation**
S. Sandrini[1], D. van Pinxteren[2], L. Giulianelli[1],  H. Herrmann[2], L. Poulain[2], M.C. Facchini[1] , S.
Gilardoni[1], M. Rinaldi[1], M. Paglione[1], B.J. Turpin[3], F. Pollini[1], N. Zanca[1], S. Decesari[1]
1 Institute for Atmospheric Sciences and Climate (ISAC), National Research Council (CNR), 40129 Bologna, Italy
2Leibniz-Institut für Troposphärenforschung (TROPOS), 04318 Leipzig, Germany
3 - Environmental Science and Engineering, Gillings School of Global Public Health, University of North Carolina at
Chapel Hill – NC 27599-7431  - USA
Correspondence to: Silvia Sandrini (S.Sandrini@isac.cnr.it)



**Abstract.** The aerosol size-segregated chemical composition was analyzed at an urban (Bologna) and a rural site (San Pietro Capofiume) in the Po Valley, Italy, during June and July 2012, to investigate sources and mechanisms of secondary aerosol formation during the summer. A significant enhancement of secondary organic and inorganic aerosol mass was observed under anticyclonic conditions with recirculation of planetary boundary layer air, but with substantial differences between the urban and the rural site. The data analysis, including a Principal Component Analysis (PCA) on the size-resolved dataset of chemical concentrations, indicated that the photochemical oxidation of inorganic and organic gaseous precursors was an important mechanism of secondary aerosol formation at both sites. In addition at the rural site a second formation process, explaining the largest fraction (22%) of the total variance, was active at night-time, especially under stagnant conditions. Nocturnal chemistry in the rural Po Valley was associated with the formation of ammonium nitrate in large accumulation mode (0.42 – 1.2 µm) aerosols favored by local thermodynamic conditions (higher relative humidity and lower temperature compared to the urban site). Nocturnal concentrations of fine nitrate were, in fact, on average five times higher at the rural site than in Bologna. The water uptake by this highly hygroscopic compound under high RH conditions provided the medium for increased nocturnal aerosol uptake of water soluble organic gases and possibly also for aqueous chemistry, as revealed by the shifting of peak concentrations of secondary compounds (WSOC and sulfate) toward the large accumulation mode (0.42 – 1.2 µm). Contrarily, the diurnal production of WSOC (proxy for secondary organic aerosol) by photochemistry was similar at the two sites but mostly affected the small accumulation mode of particles (0.14 – 0.42 µm) in Bologna, while a shift to larger accumulation mode was observed at the rural site. A significant increment in carbonaceous aerosol concentration (for both WSOC and WINC) at the urban site was recorded mainly in the size range 0.05-0.14 µm indicating a direct influence of traffic emissions on the mass concentrations of quasi-ultrafine particles.

## 1. Introduction

The knowledge of the size-segregated chemical composition of atmospheric aerosols, i.e. the chemical composition as a function of particle size, is key to the understanding of several important characteristics of particles such as optical properties, hygroscopicity and reactivity, which affect the atmospheric radiation budget, cloud formation and human health. Moreover, the size distribution of inorganic and organic components reflects their origin, hence encloses a wealth of information about aerosol formation mechanisms and atmospheric processing, including secondary formation (Seinfeld and Pandis, 1998).

Accumulation mode particulate matter mostly originates from aerosol accretion via gas-to-particle conversion of oxidized vapors (Seinfeld and Pandis, 1998). However, accumulation mode particles can be selectively scavenged by clouds and fogs which, in absence of precipitation, produce chemically-processed particles upon evaporation of water (Meng and Seinfeld, 1994). The resulting size distribution is bimodal with the small mode accounting for secondary aerosols produced uniquely by gas-to-particle conversion ("condensation mode") and the large mode containing particles that underwent cloud processing ("droplet mode") (Hering and Friedlander, 1982; John et al., 1990). Therefore, knowledge about the concentrations of aerosol organic and inorganic compounds in size-segregated aerosol samples provides information on the nature of secondary formation processes. Measurements of the size-segregated chemical composition of aerosols are traditionally performed using multi-stage impactors followed by off-line chemical analysis. Recently, the development of online mass spectrometric techniques offered new opportunities for size-segregated chemical observations with a much greater time-resolution with respect to impactors (Jimenez et al., 2003).





However, aerosol mass spectrometers (AMS) generally suffer of poor sampling efficiency for particles larger than 1
μm, and of poor sensitivity for thermally refractory compounds. Therefore, multistage impactors are still unsurpassed in
terms of the number of chemical determinations that can be performed on the samples and the range of particle sizes
that can be probed. An example of comparison of aerosol chemical measurements performed using a five-stage
impactor and AMS is provided by our previous study focusing on the 2009 field campaign in the Po Valley, Italy
(Decesari et al., 2014).
The Po Valley is a region in Europe characterized by high levels of pollution, due to the concurrent high density of
anthropogenic sources and its orographic and meteorological characteristics particularly unfavourable for pollutant
dispersion. In particular, several studies have shown how this area is dominated by secondary material during the
summer (Crosier et al., 2007), with a large presence of secondary inorganic aerosols (SIA). On an annual average, SIA
accounts for 40% of $PM_{10}$ mass at the urban site of Bologna (Matta et al., 2003; Putaud et al., 2010) while a 50%
contribution has been evaluated at the rural site of San Pietro Capofiume over shorter observation periods during winter
and summer (Carbone et al., 2010). Crosier et al., (2007) observed that most of the time, during the summer, aerosol
over the Po Valley was composed of regional ammonium sulphate and organic material, while under anticyclonic
conditions, with recirculation of air over the region, the composition was dominated by ammonium nitrate close to
ammonia emission sources.
The present study focuses on the chemical size-resolved composition of aerosol determined by two 5-stage low-pressure
Berner impactors during a field campaign performed in the summer 2012 in the Po Valley. With respect to the previous
campaigns in this area, this is the first experiment where a two-site approach was used. One-month long intensive
observations were performed at an urban site (Bologna, BO) and at a nearby rural site (San Pietro Capofiume, SPC).
The goal of this paper is to provide insights into factors controlling the variability of aerosol composition and to explore
possible formation pathways of secondary compounds in this region during the summer under different meteorological
conditions and air mass history. The two-site approach was adopted to estimate the contribution and composition of
rural background particles with respect to the urban contribution, according to the Lenschow perspective (Lenschow et
al., 2001).
Source attribution was addressed through the analysis of the time series of the main aerosol species together with
meteorological parameters, and by statistical methods such as principal component analysis (PCA), applied to the
different size classes as independent variables. The study also takes into account the comparison between an urban and
a rural site to assess the impact of traffic and other urban sources on the regional background, to explore differences in
secondary aerosol formation resulting from different meteorological conditions, and to assess the regional and local
variability of secondary aerosol formation processes. Finally, the fact that sampling was performed separately during
day and night, allowed analysis of the concentrations of aerosol constituents together with the dynamics of the boundary
layer (Gietl et al., 2008).

**2. Experimental**
**2.1 Sampling sites**
Size-segregated aerosol sampling was performed during the PEGASOS field campaign in the Po Valley (Italy), from
June 12 to July 9, 2012, at the urban site of Bologna (BO - 44° 29' N, 11° 20' E, 54 m a.s.l.) and at the rural site of San
Pietro Capofiume (SPC - 44°39' N, 11° 37' E, 11 m a.s.l.), 30 km northeast from the city of Bologna. Both sites are
located in the eastern part of the Po Valley (Fig. 1). Bologna is a city of 400.000 inhabitants, the most populous in the



southern Po Valley, with a surrounding area characterized by widespread agricultural and industrial activities and by the
presence of several high-traffic roads. Sampling was performed in the northern outskirts of Bologna, on the roof of the
Institute of Atmospheric Sciences and Climate of the National Research Council, at about 20 m above ground. San
Pietro Capofiume is a rural site characterized by a flat terrain and by croplands extending in all directions, and can be
considered an ideal receptor site for regional-scale air pollution in the Po Valley. In SPC aerosol samplers were
positioned on a platform at about 8 m above ground level.
**2.2   Aerosol sampling**
Two five-stage Berner impactors  (flow rate 80 L min$^{-1}$) with 50% particle cutpoints at 0.14, 0.42, 1.2, 3.5, and 10 μm
aerodynamic diameter ($D_p$) were used at the two sites. The particles were collected on aluminium and Tedlar foils.
Aluminium foils for carbonaceous aerosol analysis were placed on each stage of the impactor with Tedlar half foils for
ion-chromatographic analysis placed on top of them, covering 50% of the aluminium substrates (Matta et al., 2003). A
12-hour time resolution was adopted for sampling, one night-time and one daytime sample collected every day, from
21:00 to 09:00 LT and from 09:00 to 21:00 LT respectively. However, it is worth noting that nocturnal sampling
actually included several hours of light: from dawn (which in this period of the year occurs at approximately 5:30 am)
to 9 am.
**2.3   Analytical measurements**
The Tedlar substrates were extracted in 10 mL of mQ water for 30 minutes in an ultrasonic bath. The extracts were
analysed by ion-chromatography for the quantification of water-soluble inorganic species and organic acids (acetate,
formate, methansulfonate, oxalate). A TOC-VCPH analyzer (Shimadzu, Japan) was used for the determination of
water-soluble organic carbon (WSOC).
Fractions of the aluminium foils were used for the quantification of total carbon (TC) by evolved gas analysis with a
Multi N/C2100 analyser (Analytik Jena, Germany) equipped with a module for solid samples. Portions of the
aluminium foils were exposed to increasing temperature (up to 950°C) in pure oxygen carrier gas. Under these
conditions all carbonaceous matter (organic, carbonate and elemental carbon) is converted to $CO_2$ (Gelencser et al.,
2000) and TC is measured as total evolved $CO_2$ by a non-dispersive infrared (NDIR) analyser.
Aluminium foils in SPC were also used for the gravimetric determination of aerosol mass by weighing the substrates
before and after sampling on a UMT-2 microbalance with a reading precision of 0.1 μg and a standard deviation of ca.

30    1%.

**2.4   Back-trajectory calculation**
Air mass back trajectories are a useful tool when studying the aerosol composition as a function of the airmass history.
For every hour during each sampling interval (12 hours during night and 12 hours during day) 96 h back trajectories
arriving at 500 m a.g.l. were calculated by the HYSPLIT model (HYbrid Single-Particle Lagrangian Integrated
Trajectory, version 4) (Draxler and Rolph, 2003) in the ensemble mode, using input field from the global 1° GDAS





archive (http:www.arl.noaa.gov/ss/transport/archives.html). Ensembles of 27 trajectories for a given starting time for all
possible off-sets in X, Y, and Z dimensions (ca. 250 m off-set in Z, one grid cell off-set in X and Y) were calculated for
every hour, resulting in a total of 324 trajectories describing the air-mass history of each 12h sample. In addition to the
endpoints of the trajectories other HYSPLIT output parameters (sunflux, mixing layer depth) were stored and averaged
along the trajectories (sunflux) and at the receptor site (sunflux and mixing layer depth) during the sampling intervals.
Even though the mixing layer heigth provided by the HYSPLIT model might be quite inaccurate, the trend of this
parameter was taken as a proxy of the boundary layer dynamics for the campaign. In addition, residence time indices
(RTIs) were calculated by GIS analysis, reflecting the time the sampled air masses resided above certain land cover
categories (water, natural vegetation, agricultural lands, bare areas and urban areas). Details of this method are given in
van Pinxteren et al., (2010).

### 2.5  Aerosol Liquid Water Content calculation

Hourly aerosol liquid water content (ALWC) was calculated with the online versions of the Extended Aerosol Inorganic
Model III (E-AIM, http://www. aim.env.uea.ac.uk/aim/aim.php; (Clegg et al., 1998); (Wexler and Clegg, 2002). The
inorganic concentrations for sulfate, ammonium, nitrate, sodium and chloride measured by two HR-TOF-AMS, placed
respectively in BO and in SPC during the campaign, were used as inputs in conjunction with RH, while temperature
was kept fixed at 298.15 by this model. The size segregated concentrations of the inorganic components collected by
the Berner impactors could not be used for this purpose, due to their low temporal resolution (12 hours), resulting in a
flattening of RH and T values averaged over the sampling time, which did not justify the presence of ALW on particles.
Particulate water retained by polar organic matter was neglected in the calculations, because the inclusion of ionic
organic compounds (oxalic, glutaric and maleic carboxylic acids) had been shown to play only a minor role in water
uptake during the campaign (Hodas et al., 2014). The ALWC calculated from AMS relates to the fine fraction (< 1 µm
diameter) of aerosol.

### 2.6  Principal Component Analysis

Source categories for the 5 impactor size intervals of particulate matter were studied by means of Principal Component
Factor Analysis (PCA) using the XLStat software (Addinsoft, version 2013.2.04). Besides the concentrations of particle
constituents chloride, sulfate, nitrate, sodium, ammonium, magnesium, calcium and WSOC, the database for PCA also
included the modelled meteorological parameters from the HYSPLIT model listed in the previous section, the measured
meteorological parameters temperature and relative humidity and the modelled residence times from GIS analysis.
The values below detection limits were replaced by half the respective value of detection limit (Farnham et al., 2002) in
the final dataset.
Since a prerequisite of PCA is the normal distribution of the variables used in the analysis, the normal distribution of
concentration data has been checked by the Shapiro-Wilk normality test. Data that were not normally distributed were
log-10 transformed before the analysis.
The orthogonal transformation method with Varimax rotation of Principal Components was applied to redistribute the
variance in order to generate more interpretable factor loadings and scores (Vandeginste, 1998). As there are no defined
criteria for the number of factors which are used in the Varimax rotation, we performed several PCAs with varying
numbers of rotated factors (4–9) and judged on the interpretability of the results by trying to assign a physical meaning





to the extracted factors. The number of rotated factors was regarded too high if factors showed very low contribution to
the overall variance and no distinct physical meaning. Contrarily the number was regarded too low if previously
resolved sources were now folded into one principal component. The most reasonable results were obtained by rotating
the first 6 factors.
**3.    Results and discussion**
**3.1 Back trajectory patterns**
The PEGASOS summer campaign was characterized by the occurrence of different meteorological patterns, with a first
part characterized by days of very perturbed weather followed by stable anticyclonic conditions, and a second part
experiencing more variable meteorological conditions. An overview of the main air mass transport patterns intersecting
the area during the campaign, from back-trajectory analysis, is reported in a parallel paper (Decesari et al., in
preparation). In brief, a hierarchical cluster analysis (Dorling et al., 1992) of the obtained back trajectories was
performed to reduce the number of "origins of air masses", appointing every calculated individual back trajectory to the
most appropriate cluster, roughly corresponding to a specific synoptic situation. At each step of the process, the
appropriate number of clusters was selected by looking at the variations of the total spatial variance (TSV - defined as
the sum of the squared distances between the endpoints of the single trajectory and the mean of the trajectories in that
cluster). The optimum number of clusters was selected in correspondence with the number after which the TSV did not
vary substantially. The analysis led to identification of five main patterns affecting the Po Valley during the
experimental campaign. A map with the mean trajectories for each cluster and the corresponding percentage of
occurrence for trajectories calculated over 96 h and arriving at 500 m a.g.l.is shown in Fig. 2.
Clusters 1 and 5, defined respectively "WEST low" and "EAST low" according to their low travelling altitude (below
1000 m a.g.l.) and prevalent direction, were characterized by short trajectory lengths, and correspond to a higher
residence time of air masses in the basin. Cluster 1, in particular, had the highest occurrence and accounted for 42% of
the total trajectories. Both included a smaller number of very short trajectories which were defined respectively as
"WEST-low local" and "EAST-low local", occurring during the days characterized by stagnant conditions and low
wind speed, and were associated with the accumulation of pollutants.
**3.2 Bulk PM10 aerosol composition**
The size cut between fine and coarse particles in the Berner impactor size distributions is set to a 1.2 μm aerodynamic
diameter (i.e., the size cut between the $3^{rd}$ and the $4^{th}$ impactor stage), therefore in this study $PM_{1.2}$ and $PM_{1.2-10}$
represent the fine and the coarse aerosol fractions, respectively. Fig. 3 shows the time series of $PM_{1.2}$ and $PM_{1.2-10}$ mass
concentrations and of the contribution of $PM_{1.2}$ and $PM_{1.2-10}$ for the rural site SPC, together with air mass categories and
back trajectory length. The aerosol mass was not available for BO. Air mass categories indicate the prevalent direction
of airmasses during each sampling day.
During the campaign, the days showing the lowest aerosol mass concentrations were characterized by the longest
trajectories, which corresponded to air masses transported over long-range from the North Atlantic Ocean (WEST), at
the beginning and at the end of the sampling period. The aerosol mass increased from 15 to 18 June, together with a
sharp decrease of the trajectories length, following the onset of an anticyclonic period with low wind and air stagnation



over the Po Valley. The $PM_{1.2}$ to $PM_{10}$ ratio increased correspondingly during such days, suggesting a more important
contribution of secondary aerosol. An episode of Saharan dust transport was observed during a period of air transport
from south, starting on 19 June at 5 km height (Bucci et al., in preparation) with maximum on 20 June when it reached
the PBL, resulting in an increase of $PM_{10}$ mass at ground level. The highest contributions of $PM_{1.2}$ to $PM_{10}$ were
observed most of the times when 4-day trajectories were very short (< 1500 km). During the first, persistent stagnation
period, lasting from 16 to 19 June, the $PM_{1.2}$ contribution to $PM_{10}$ was the highest, with maxima during the night,
peaking at 67% of total $PM_{10}$ mass on 17 June.
Table 1 lists the concentrations of the aerosol chemical constituents, separately for $PM_{1.2}$ and $PM_{1.2-10}$ fractions and for
day and night samples. All the chemical species concentrations were higher at the urban compared to the rural site but
with small differences in most of the cases. The significantly higher concentrations of fine TC in BO can result from a
higher contribution of elemental carbon (EC) and organic carbon (OC) from vehicular traffic at the urban site. Coarse-
mode $Mg^{2+}$ and $Ca^{2+}$ also occurred at higher concentrations at the urban site, suggesting a source from road dust
resuspension for these mineral elements. The only exception was represented by the nocturnal concentrations of fine-
mode nitrate, on average five times higher at SPC compared to BO, and so for the counter-ion ammonium.
The concentrations of many species were higher at night compared to daytime (Tab. 1). The nocturnal enhancement can
be due either to the accumulation of pollutants in a shallow boundary layer or to enhanced formation/emission of
specific aerosol species at night-time. Fig. 4 shows the average $PM_{10}$ composition at the urban and at the rural site
separately for day and night, with indication of the percentage contributions of each species on the total mass of the
measured compounds. The change in $PM_{10}$ chemical composition between day and night at the rural site indicates that
the enhanced mean concentrations found at night were not purely an effect of atmospheric dynamics but were impacted
by chemical processes that led to the formation of specific aerosol compounds, especially ammonium nitrate, in the dark
hours of the day.
Water insoluble carbon (WINC) was calculated as the difference between TC and WSOC (WINC = TC − WSOC)
(Matta et al., 2003). In order to obtain the mass of water-soluble organic material (WSOM) and water insoluble
carbonaceous material (WINCM) conversion factors were applied to include elements different from carbon in the
organic molecules. Two distinct factors were used respectively for the soluble and the insoluble fraction of organic
carbon. WSOC was multiplied by 1.9 at the rural site and by 1.7 at the urban site, applying the two factors derived from
AMS measurements performed at the two sites during the same campaign (Gilardoni et al., 2014). WINC was
multiplied by 1.2 as found in the literature (Zappoli et al., 1999).
The total mass concentration of the chemical species determined on impactor samples averaged 12.2 µg m$^{-3}$ at the urban
and 8.2 µg m$^{-3}$ at the rural site during daytime and 15.1 and 15.9 at night-time, respectively. It is interesting to note how
in daytime the aerosol loading was higher at the urban compared to the rural site, indicating a higher contribution from
urban sources, though the chemical composition was to some extent homogeneous. By contrast at night-time the aerosol
mass was similar at the two sites but the chemical composition was different, with an enrichment of ammonium nitrate
at the rural site.
The measured mass of $PM_{10}$ at the urban site consisted on average of 42% and 40% secondary inorganic aerosol (SIA)
in daytime and night-time respectively. At the rural site, SIA represented 46% in daytime and 50% at night, due to a
higher contribution from fine-mode ammonium nitrate. Carbonaceous matter was the dominant fraction at the urban site
with 48% and 49% of the measured mass in daytime and night-time, respectively, and with the soluble fraction
accounting for 26% and 27% of the mass in the two cases.





Finally, it is worth to remind that the above results are sensitive to sampling artifacts which can affect aerosol collection
with low-pressure impactors (e.g., evaporative losses of semivolatile compounds). Figures S7 and S8 report the
comparison between SIA measurements with the Berner impactor at SPC and co-located measurements using a
different off-line system (a HiVol sampler) and an online method (HR-ToF-AMS). The results show that the Berner
impactor observations are generally in line with the parallel measurements and especially that the main features of the
time trend (e.g., the sharp diurnal variations in nitrate concentrations) are reproduced by all the instruments.

### 3.3 Size-resolved aerosol composition

The time-series of size-resolved sulfate, nitrate and WSOC concentrations are shown in Fig. 5 (similar plots for
additional chemical components are reported in the Supplementary Material, Figg. S1-S6). The figure highlights
significant differences between the urban and the rural site in the formation of secondary inorganic and organic aerosol,
especially during the two periods of stagnant conditions, i.e. 16-19 June and 5-7 July, which favored the accumulation
of aerosol compounds from local sources. During the first of such events, the sulfate concentrations increased in BO
and SPC to a similar extent in daytime, while higher concentrations were measured in SPC at night. Fine-mode nitrate,
consisting primarily in ammonium nitrate, was virtually absent during the day throughout the campaign, particularly in
BO, as a consequence of the high summer temperatures, which favored the thermal decomposition of $NH_4NO_3$ into gas
phase ammonia and nitric acid. Higher fine-mode nitrate concentrations were instead measured in SPC, reaching high
levels at night. For both sulfate and nitrate the most prominent enhancements affected the accumulation mode, i.e. size
bins 2 (0.14 – 0.42 μm) and especially 3 (0.42 – 1.2 μm) of the impactor, which correspond to small and large
accumulation mode.
Aerosol WSOC exhibited higher concentrations under stagnant conditions similarly to SIAs. During the first stagnation
event, in particular, higher nocturnal concentrations in the accumulation mode were measured in SPC compared to BO
(Fig. 5). Conversely, WSOC concentrations were in general higher in BO than in SPC in the quasi-ultrafine (0.05 – 0.14
μm) fraction, with an average concentration of 0.39 μg m$^{-3}$ (17% of the total WSOC in the PM$_{10}$ fraction) compared to
0.19 μg m$^{-3}$ (11% of total WSOC in PM$_{10}$). Since secondary formation is believed to represent the major source of
WSOC in the absence of biomass burning (Weber et al., 2007), the quasi-ultrafine WSOC excess at the urban site could
result from the increased condensation of secondary products on the large surface area of a higher number of very small
particles found at the urban site. In addition to freshly nucleated particles, in fact, this aerosol fraction, twice as high in
number at urban sites compared to rural areas (Westerdahl et al., 2005), includes particles released by anthropogenic
sources such as combustion emissions from vehicular traffic. In addition, this quasi-ultrafine mode excess of WSOC
mass could also reflect a direct contribution from anthropogenic primary emissions (Zhang et al., 2012).
On 5 and 6 July, the Po Valley was again influenced by stagnant conditions, with low wind speeds and short air-mass
trajectories. However, the stagnation event in early July was considerably shorter than that of 16–19 June. During such
event, high concentrations of fine-mode sulfate were simultaneously observed in BO and SPC with the 6 July
experiencing the highest diurnal concentration of the whole campaign (4.9 μg m$^{-3}$ in PM$_{10}$ in BO). Nitrate similarly
increased at both sites at night, persisting in daytime in BO only, and reaching the diurnal maximum of 4.2 μg m$^{-3}$ on
PM$_{10}$. Similarly, WSOC increased at both sites, though more significantly in BO, where it reached the highest average
diurnal concentration on 6 July. This day was characterized by the highest daytime relative humidity (47.8% and 60.4%



in BO and SPC respectively, averaged over the time span of Berner impactor samplings) and a relatively lower
temperature (27°C at both sites). On that day, scattered clouds were observed near the sampling stations and rainfall
occurred in the northern sector of the Po Valley. Such conditions apparently favored the formation of secondary organic
(WSOC) and inorganic aerosol compounds in the Po Valley basin.
Fig. 6 shows the size-segregated concentrations of sulfate, nitrate, ammonium and WSOC during one day characterized
by background conditions (15 June) and one day characterized by stagnant conditions (18/6), separately for diurnal and
nocturnal hours. On the background day (panels a and c), fine particles in BO exhibited a maximum in the speciated
aerosol mass in the small accumulation mode (0.14 – 0.42μm), with a relative increase in the large accumulation mode
(0.42 – 1.2μm) at night. The speciated aerosol mass in SPC was almost evenly distributed between the two modes at
night and day. Nitrate was always present in the coarse mode, where non-volatile nitrate salts can form through reaction
of gaseous nitric acid with alkaline soil particles or resuspended dust (Harrison and Pio, 1983; Laskin et al., 2005).
Under stagnant conditions (panels b and d) the speciated particle mass concentrations increased but the peak in the size
distribution, which for BO was again observed in the small accumulation mode, shifted to the large accumulation mode
in SPC both in daytime and at night. The nocturnal increase of large accumulation mode particulate matter
concentration was much more evident at the rural site (SPC) than at the urban site (BO), mostly because of an increase
of ammonium nitrate (showing more than 3 times higher concentrations with respect to background conditions), but
also accompanied by increases in WSOC and sulfate concentrations. The SIA mass (i.e. the sum of sulfate, nitrate and
ammonium) reached 67% of the total $PM_{1.2}$ mass in the night of 16 June, during the first event of stagnation. The
increase of SIA (and, to a lesser extent, of WSOC) in large accumulation mode aerosols under stagnant conditions was
therefore the main process modulating the impactor size distribution of submicron aerosol during the experiment. Our
data seem to exclude that this shift in diameter of SIA-containing aerosols from the small to the large accumulation
mode size range is due to impactor sampling artifacts (see Supplementary material SI-2, Figure S9). Under the
hypothesis that large accumulation mode aerosols correspond to droplet mode particles (Hering and Friedlander, 1982;
John et al., 1990; Meng and Seinfeld, 1994), the formation of secondary aerosols through aqueous-phase chemical
reactions must play a crucial role in accumulation mode aerosol growth in this environment.

### 3.4 Secondary aerosol formation under stagnant conditions

#### 3.4.1 Secondary inorganic components

The processes responsible for the accumulation of sulfate, nitrate and secondary organic aerosol (SOA) on stagnation
days were further investigated by analyzing the relationship between the (size-segregated) concentrations of secondary
inorganic compounds and relative humidity (RH), as well as with aerosol liquid water content (ALWC). SPC was
characterized by higher levels of RH than Bologna, especially at night. During the campaign RH and temperature
profiles at the two sites showed substantial day-night variations as a consequence of diabatic processes at the surface
(nocturnal radiative cooling versus daytime heating from solar irradiation). As a consequence, the highest RH occurred
at both sites just before dawn, from 4 to 5 am, and the minimum in the afternoon from 1 to 3 pm. Upon averaging
hourly RH data over the Berner impactor sampling time, a mean RH value of 32% was obtained for BO (min 15%, max
66%) for daytime sampling periods and 48% (min 24%, max 80%) for night-time conditions. RH averaged 40% in SPC
in daytime (min 19%, max 73%) and 69% at night (min 40%, max 92%).



Since the equilibrium constant for the reaction of $NH_4NO_3$ formation is both RH and temperature dependent (Stelson
and Seinfeld, 1982), thermodynamic conditions were more favorable in SPC than in BO for the existence of condensed
phase nitrate. Both temperature and RH affect the equilibrium of $NH_4NO_3$, but the changes in RH were more marked. In
particular, nocturnal periods of time with RH greater than the relative humidity of deliquescence (RHD) of hygroscopic
salts (82% for ammonium sulphate and 62% for ammonium nitrate at 25°C (Watson et al., 1994)) were considerably
longer in SPC than in BO, and therefore deliquesced particles were common at the rural and very rare at the urban site.
This was confirmed by the simultaneous enhancements of the aerosol liquid water content (ALWC), calculated from
hourly averaged AMS data at the two sites (see the experimental section) by the E-AIM model (Fig. 7). While the
ALWC in SPC showed a consistent average diurnal trend, with a maximum before dawn when both RH and the
concentrations of hygroscopic salts were the highest (Hodas et al., 2014), the diurnal variation in Bologna was less
marked, and night-time ALWC concentrations were one order of magnitude smaller than at the rural station (Fig. 7).
Considering the Berner impactor sampling periods, ALWC averaged 0.33 µg m$^{-3}$ in daytime in BO (min 0, max 11.2 µg
m$^{-3}$) and 0.76 µg m$^{-3}$ at night (min 0, max 16.6 µg m$^{-3}$), while in SPC it averaged 0.21 µg m$^{-3}$ during day (min 0, max
10.4 µg m$^{-3}$), and 6.55 µg m$^{-3}$ during night conditions (min 0, max 59.3 µg m$^{-3}$). The 12-h averaged ALWC data shows
concentrations increasing rapidly for RH above 60% (Figure 8). There is a considerable variability in ALWC levels that
must be attributed to the availability of hygroscopic material in the aerosol, as particulate water in subsaturated
condition is a function not only of RH but also of the molar concentration of dissolved material in the aerosol.
Figure 8 (panels b,c) shows the relation between nitrate and sulfate concentrations separately in the small accumulation
mode (condensation mode, CD)  and in the large accumulation mode (droplet mode, DL)  versus RH for the two sites.
The correlation between particulate nitrate concentration and RH was strong for both modes in SPC where ALWC
levels above 1 µg m$^{-3}$ were frequent.
Our data are in agreement with the findings of Hodas et al., (2014) indicating that particulate nitrate was the primary
driver of ALWC observed at night in the rural Po Valley in summertime. The same study suggested that ALWC
enhanced the particle phase partitioning of water soluble organic gases and provided a medium for aqueous phase
organic reactions that can form SOA. More generally, ALWC can favor – and sometimes can be necessary for – the
formation of SOA and SIA in large accumulation mode aerosols (the droplet mode). Figure 8c shows that no
relationship was observed between sulfate and the local estimates of ALWC at either site, with the exception of a
moderate positive correlation for the droplet mode in SPC but not in BO. These results indicate that most sulfate was
formed at the regional scale during this study (through cloud chemistry in the larger accumulation mode or gas phase
chemistry in the smaller accumulation mode). However a small but non-zero increment of droplet-mode sulfate
concentration in SPC can be attributed to the greater ALWC characterizing the rural site with respect to the urban site.
As a final remark, in this discussion, we attributed the high concentrations of ammonium nitrate at the rural site to the
more favorable thermodynamic conditions respect to the urban site. This process was amplified by a positive feedback
of ammonium nitrate itself that, by increasing the ALWC in the aerosol phase, further promotes the uptake of
precursors ($NH_3$, $HNO_3$, $N_2O_5$) from the gas-phase. We do not consider here the effect of the different distribution of
gaseous precursors of ammonium nitrate in the Po Valley, especially of ammonia which is enriched in the rural areas
(Sullivan et al., 2015), which will be object of a future study.



**3.4.2 Carbonaceous species**
The distribution of the carbonaceous components of aerosol (WSOC, WINC = TC-WSOC) over the Berner impactor
size intervals, with emphasis on accumulation mode particles (condensation and droplet modes) is hereby discussed.
The dependence of WSOC on aerosol liquid water content (ALWC) is also investigated analogously to the case of SIA
treated in the previous section.
$PM_{10}$ WSOC concentrations in this study ranged from 0.13 to 4.6 µg C $m^{-3}$ (average: 2.1 µg C $m^{-3}$) in BO and from 0.28
to 5.2 µg C $m^{-3}$ (average: 1.6 µg C $m^{-3}$) in SPC. On average, 79% and 77% of this $PM_{10}$ WSOC concentration was in
the fine ($PM_{1.2}$) fraction in BO and SPC respectively, with only slightly higher contributions at night than in daytime at
both sites.
The water-soluble fraction of total carbon (WSOC/TC) in $PM_{1.2}$ was 52% on average in BO and 61% in SPC. The
WSOC fraction in SPC was comparable to what observed at other rural sites, e.g. 57% in $PM_1$ in K-puszta (Hungary)
during the summer (Krivacsy et al., 2001). The greater water-soluble fraction of carbon found at SPC with respect to
BO is in line with literature results, showing higher WSOC fractions in rural areas as a consequence of the concurrent
higher input of SOA and the reduced fraction of insoluble carbonaceous particles from traffic sources (Weber et al.,

15    2007).

The left panels in Fig. 9 show the linear regressions of WSOC versus TC in submicron particles (impactor stages 1 to 3)
at the two sites, separately for day and night conditions. At the urban site, a higher slope was observed in daytime (0.71
± 0.10 at 95% confidence level) than at night-time (0.57 ± 0.08), pointing to the effect of a daytime source for WSOC
(consistent with photochemical SOA formation). In daytime, the WSOC fraction of TC in BO overlaps well with that
observed in SPC (0.72 ± 0.05). Contrary to BO, however, WSOC fractions in SPC were similar between night and day
(0.70 ± 0.07 at night). Fig. 9 shows, in fact, that WSOC and TC occurred in similar proportions in daytime between
SPC and BO, but with smaller concentrations in SPC (which were therefore "diluted" with respect to BO). In addition,
carbonaceous aerosol concentrations increased at night in SPC and in similar proportions between WSOC and WINC,
hence producing an aerosol with different characteristics in SPC with respect to BO.
The correlation of WSOC with a non-volatile SIA component (sulfate) in $PM_{1.2}$ is shown in the right panels of Fig.9. A
good correlation ($R^2 = 0.7$) was observed at both stations in daytime, suggesting that WSOC shared a photochemical
source with sulfate. The correlation in BO was much smaller at night ($R^2 = 0.4$) than in daytime, which is expected,
because particulate organic compounds have multiple sources other than photochemistry. Interestingly, the correlation
between WSOC and sulfate remained high (0.7) at night in SPC, pointing to a common nocturnal source for WSOC and
sulfate at the rural site.
Fig. 10 shows, for the two sites, the size-resolved concentration time series of WSOC and WINC for quasi-ultrafine
mode (0.05 – 0.14 µm ), condensation mode (0.14 - 0.42 µm) and droplet mode (0.42 – 1.2 µm) particles. The quasi-
ultrafine fraction provided the smallest contribution to aerosol mass, but with significantly higher concentrations at the
urban compared to the rural site for both WSOC and WINC. This feature was observed also in previous studies (Sardar
et al., 2005; Snyder et al., 2010; Zhang et al., 2012). The urban excess of WINC witnesses the effect of local emissions
of insoluble primary carbonaceous particles. On average WSOC accounted for 52% of quasi-ultrafine TC in SPC and
only 42% in BO, with the lower WSOC fraction at the urban site caused by the higher concentrations of water insoluble
carbon. An urban increment for WSOC in quasi-ultrafine particles can be observed, although smaller than for WINC,





and can be explained by local sources of fresh SOA and by condensation on a greater number of ultrafine particles. WINC in this size range displayed nocturnal excess compared to WSOC, particularly in BO, without a clear relation with the trajectory lengths. We argue that the nocturnal peaks in quasi-ultrafine WINC concentrations could be related to the morning traffic rush hours, which, during the summer, had a maximum at 8-9 am, before the break-up of the nocturnal boundary layer and were therefore included in the nocturnal sampling periods. The evening traffic rush time between 7 and 8 pm had apparently a minor effect, since occurring while the boundary layer was still well mixed.

The distribution of carbonaceous fractions in the accumulation mode (condensation and droplet modes) showed that WSOC was in general dominant over WINC, more at the rural than at the urban site. The average WSOC/TC ratio in the condensation mode was 65% in SPC and 60% in BO, while showing a greater difference between the two stations in the droplet mode (66% in SPC and 56% in BO). The WSOC concentrations were inversely related to trajectory length, indicating an effect of stagnation on oxidized organic aerosol production. Interestingly, the increase of WSOC levels during the stagnation periods did not affect the same size fractions to the same extent at the two sites. During the first episode, the maximum daytime WSOC concentration in the condensation mode, recorded on 19 June, corresponded to a 185% increase in BO and 150% in SPC compared to 15 June, the last day under the influence of North Atlantic circulation before the onset of the local recirculation. The enhancement of WSOC in daytime for droplet mode particles amounted to 150% in BO and 195% in SPC. Therefore, the accumulation of water-soluble organic compounds occurred approximately to the same extent at the two sites in daytime, but the increase was more marked in droplet mode particles in SPC and in the condensation mode in BO. The maximum nocturnal WSOC concentration, found on June 18, corresponded to an enhancement in the condensation mode of 140% and 325% in BO and SPC, respectively, with respect to the background conditions of 15 June. The same increase in the droplet mode amounted to 115% (BO) and 440% (SPC). Therefore, the change in WSOC concentrations between background and stagnating conditions was more heterogeneous between sites for nocturnal samples than for the diurnal ones, which is expected because the atmosphere is much more stratified at night and atmospheric composition at ground level is more impacted by local conditions. In this case, a nocturnal enhancement of WSOC concentrations during the first stagnation period occurred only in SPC, with a maximum in the droplet mode.

During the second stagnant period, on 5-6 July, WSOC increased considerably in daytime in BO in both size ranges, while only a small increase was observed in SPC and limited to the droplet mode.

The behavior of accumulation mode WSOC after the onset of stagnating conditions was therefore reversed during the (short) July event with respect to the first episode of 16 – 20 June, with a marked increase in the droplet mode occurring in BO in July while interesting the SPC site in June (especially at night). The days (and nights) of maximum increase of droplet mode WSOC were in fact those showing the highest submicron nitrate concentrations, and were always humid days (or nights). Fig. 8d shows that WSOC was positively correlated with ALWC only in the droplet mode and only in SPC. The lack of correlation for BO samples can be explained by the very short duration of the humid stagnation period in July. Clearly, the increase of WSOC in droplet mode aerosols in the stagnation periods was not homogeneous in the Po Valley and was associated locally to the presence of deliquesced particles. These findings indicate that the enrichment of WSOC was contributed by aqueous processes, including condensation in the aerosol liquid water, which were active preferentially during colder night-time hours. Fig. 8 c,d also shows that the behavior of WSOC reflects that of sulfate in SPC for droplet mode aerosols, which explains the good correlation between WSOC and sulfate for night-time samples observed only at the rural site where ALWC was high (Fig. 9 right panels).





**3.5 Principal component analysis**
In the previous section, we focused on the time trends and size distributions of major carbonaceous and inorganic ionic
species and we concluded that at least two secondary formation processes were active in the Po Valley: a first, probably
photochemical, active throughout the campaign at both stations affecting the concentrations of all species and
particularly in the condensation mode during stagnation periods; and a second one associated with deliquesced particles,
and selectively important for nitrate (both condensation and droplet mode) and to a lesser extent for sulfate and WSOC
in droplet mode particles. In this section, we will extract source information from all the chemical dataset and from
ancillary information. Principal component analysis (PCA) was used to analyse the variability of the main variables of
interest (the concentration of main SIAs and of WSOC and WINC) in conjunction with the variability of the
concentrations of minor species and tracers as well as of the physical parameters of the atmosphere. Six principal
components were retained for interpretation of the SPC and BO datasets, explaining respectively 79% and 77% of the
total variance. The results of the PCA are summarized in Table 2 as factor loadings, which represent the correlation of
each variable with each factor, and hence suggest possible sources, formation mechanisms and source regions. For
better clarity, loadings with absolute values below 0.2 ($|x| \leq 0.2$) were omitted and only those with absolute values
larger than 0.6 ($|x| \geq 0.6$) were considered "high" (in bold in the table).
The rotated component 5 (RC5) explained the largest fraction of the dataset variance (22%) in SPC. It involved high
loadings of nitrate and ammonium for all the size classes. Among the meteorological parameters, the factor was
strongly positively correlated with RH (0.82), and negatively with temperature (-0.85). Thus RC5 describes the local
meteorology and the night-time condensation of ammonium nitrate in large accumulation mode particles (0.42 – 1.2
μm). Ammonium nitrate condensation in SPC was moderately positively correlated with droplet mode sulfate, WSOC
and oxalate, compounds (sulfate and oxalate) that share a source in aqueous secondary formation processes. A similar
source was identified in the Bologna dataset in RC5 which, compared to SPC, explained only a smaller fraction (9%) of
the total variance. The relationship with temperature and relative humidity was weaker in BO than in SPC, though in the
same direction, and no clear relation was observed with droplet mode WSOC and sulfate
RC1 was the second most important factor in SPC, explaining 15% of the total variance in the dataset. Sulfate, oxalate,
ammonium and WSOC were the species which correlated most with this factor, especially in the small (0.14 – 0.42 μm)
and large (0.42 – 1.2 μm) accumulation mode. This factor was positively correlated with the sunflux integrated along
the airmass trajectory. This factor can therefore account for SIA and SOA photochemical production. An analogous
factor in BO (RC2) explained a similar amount of total variance (17%). A positive correlation (0.6) with relative
humidity was observed in BO, which was not significant in SPC.
The third factor in SPC is RC6, accounting for 14% of the total variance. This rotated component showed high loadings
particularly for $Ca^{2+}$ but also for $Mg^{2+}$ in size bins 3 to 5, corresponding to particles from 0.42 μm to 10 μm. Oxalate
also showed a significant loading in size bin 5 (3.5 – 10 μm), which suggested the uptake of gas-phase carboxylic acids
by mineral particles, in agreement with past observations (Laongsri and Harrison, 2013; van Pinxteren et al., 2014).
This factor is analogous to RC6 in the BO dataset, which explains 11% of the total variance, and was attributed to road
dust resuspension. High loadings in BO were shifted toward larger diameters of particles (1.2 to 10 μm).



The fourth rotated component in SPC, which explained another 12% of the total variance, displayed the highest
loadings for magnesium in coarse mode particles and sodium in size bins 3 to 5 (from 0.42 to 10 µm). Sulfate in size
bin 5 (3.5 – 10 µm) was also moderately correlated with this factor, and nitrate too, but to a lesser extent. This factor
can be interpreted as a contribution from seasalt components. The corresponding rotated component in BO was RC1,
which explained 19% of the total variance and, compared to SPC, displaied higher loadings for sulfate and nitrate in
size bin 4 (1.2 to 3.5 µm).
The fifth rotated component, explaining 8% of the total variability, only included high loadings for residence times
indexes (RTI), without any relevant correlation with other parameters. This factor was identified both in SPC and in
BO, explaining respectively 8% and 10% of the total variance. While this factor does not represent an aerosol source, it
indicates that during this campaignthe impact of air mass history (long-range transport) was likely small as compared to
other impacts such as day/night variability or local impacts.
Finally the last rotated component, which explained an additional 8% of the total variance in SPC and 11% in BO,
contained high loadings only in the first size range (0.05 – 0.14 µm) for sulfate, ammonium, and to a lesser extent
WSOC. The significant loading of sulfate and ammonium in quasi-ultrafine particles, together with a moderate positive
correlation with solar radiation and temperature, suggests a possible source for this component in ultrafine particle
nucleation. The role of these chemical species in the formation of new particles and their condensation/coagulation on
smaller particulate matter are well-known, as well as the influence of solar radiation on this aerosol generation process
(Hamed et al., 2010). The negative relation with relative humidity confirmed that this source was active during the day,
when relative humidity was minimum.
**Discussion and conclusions**
The PCA results indicate that several factors determined the variability in the size-segregated chemical composition in
the region during the PEGASOS Po Valley field campaign, but each of them affected preferentially specific size
intervals with an overall effect of shaping the aerosol mass distribution at the two sites. Two factors corresponding to
seasalt and mineral dust with absorbed nitrate regulated the concentrations and composition of coarse particles ($PM_{1.2-}$
$_{10}$), while only one factor was found to determine an enrichment of ammonium sulfate in the quasi-utrafine range.
Finally, the variability in composition of accumulation mode aerosol could be reduced to two factors, with one related
to regional-scale photochemical formation of SOA and SIA, and a second one more dependent on local conditions at
surface level and causing a nocturnal increase of SOA and SIA in the droplet mode. However, if the factor for
photochemical secondary aerosols was equally represented at the two sites, the other one (for night-time condensation)
was much more important at the rural station (SPC) than in BO. The effect on a simple non-volatile SIA component,
sulfate, is exemplified in the scatter plot in Fig. 11. A good correlation ($R^2 = 0.9$) was indeed observed between the
accumulation mode sulfate concentrations at the two sites in daytime (Fig. 11), with only a slight dilution (-12%) of the
concentrations at the background site with respect to the urban site, which is expected for an aerosol component which
is typically associated to regional-scale photochemical pollution. By contrast, the correlation between the concentration
trends at the two sites is much lower at night, especially as regards the droplet mode, where significantly higher sulfate
concentrations occurred in SPC compared to BO during the stagnant days from 16 to 20 June. Clearly, stagnating
conditions and the onset of thermal inversions at night favored a partial "chemical segregation" of air masses in the



surface atmospheric layers within the Po Valley, and the size-segregated chemical composition evolved separately at
the urban sites close to the Apennine foothills with respect to the rural areas in the inner Po Valley during dark hours.
Specifically, rural areas were characterized by the presence of ammonium nitrate and by ALWC levels above 10 $\mu$g m$^{-3}$.
Deliquesced aerosols could host aqueous phase formation of sulfate (via reaction of $SO_2$ with $H_2O_2$).
Similarly to SIAs in the droplet mode, water-soluble products of volatile organic compound (VOC) oxidation could
readily be taken up by deliquesced particles in SPC at night. A meaningful fraction of the newly formed (1–3 h old)
WSOC mass, in fact, has been shown to possess similar semi-volatile properties to $NH_4NO_3$ (Hennigan et al., 2008;
Wilson et al., 2006) and can rapidly partition to aerosol water or cloud/fog droplets. The nature of the nocturnal
enrichment of WSOC in the droplet mode, depending on the reactivity in the aqueous phase, can be described by either
a reversibile mechanism (condensation of water soluble organic compounds triggered by the change in RH and ALWC)
or an irreversible reaction (oxidation of VOCs or OVOCs with production of stable compounds). Hodas et al. (2014),
based on measurements performed during the same campaign in SPC, observed an exponential decrease in gas phase
glyoxal concentrations with increasing ALW, and a local nocturnal production of aqueous SOA was indeed observed by
parallel near real-time WSOC sampled with a Particle into Liquid Sampler (PILS) at the same site (Sullivan et al.,
2015). The analysis of impactor samples provides only a few clues to disentangle the two effects. The only two organic
markers for SOA that were quantitatively determined in all samples were oxalate and methanesulfonate (MSA). The
robust correlation between WSOC, oxalate and sulfate, both in daytime and at night-time in SPC, indicates that the
accumulation of particulate polar organic compounds contributed to the (irreversible) production of stable (oxidized)
species. Oxalate is generally known to share with sulfate an important aqueous-phase oxidation pathway (Sorooshian et
al., 2006). MSA is a more specific marker than oxalate, being related to the atmospheric processing of dimethylsulfide
(DMS) whose emissions are unevenly distributed on the Earth surface, and can be intense in biogenically-rich marine
waters. Our data showed indeed that fine-mode MSA was maximum in the days between 26 June and 1 July,
characterized by an easterly or a south-southwesterly circulation, bringing marine air masses into the Po Valley basin.
Such increase from long-range transport affected particularly the size intervals 0.05 – 0.14 $\mu$m and 0.14 – 0.42 $\mu$m.
MSA concentrations in the droplet mode (0.42 – 1.2 $\mu$m) showed instead an enhancement at night under stagnant
conditions, similarly to ammonium nitrate (see Supplementary Material), particularly marked in SPC during both
episodes and in BO only during 5-6 July. The increase of MSA in droplet mode particles under stagnating conditions
points to a DMS (or other reduced sulfur species) source other than from the marine boundary layer. In inland areas
DMS has sometimes been reported as dominantly from terrestrial sources (vegetation and soils) and anthropogenic
sources (manure and livestocks), with higher temperatures and solar radiation enhancing its emission (Perraud et al.,
2015). These findings suggest that the VOCs participating to the formation of WSOC in the Po Valley also included
organic compounds emitted by agricultural activities or even by natural sources, and that ALWC in the atmospheric
nocturnal surface layer acted as a medium for their formation during summer time.

**Acknowledgements**
This research was conducted as part of the "Supersito" Project, supported by Emilia Romagna Region and Regional
Agency for Prevention and Environment (ARPA Emilia Romagna) under Deliberation Regional Government n. 428/10.
The work was also made possible by the European Commission under the Framework Programme 7 (FP7) projects
PEGASOS (Grant Agreement 265148) and BACCHUS (Grant Agreement 603445), which are highly acknowledged.





1     **Table 1 – Fine (PM$_{1.2}$), coarse (PM$_{1.2-10}$) and PM$_{10}$ mean concentrations (in µg m$^{-3}$, from integrated impactor**
2     **mass size-distributions) for main inorganic ions, water-soluble organic carbon (WSOC) and total carbon (TC)**
3     **separately for day (D) and night (N) samples during the PEGASOS summer campaign.**

| | PM$_{1.2}$ | | | | PM$_{1.2-10}$ | | | | PM$_{10}$ | | | |
|---|---|---|---|---|---|---|---|---|---|---|---|---|
| | BO (urban) | | SPC (rural) | | BO (urban) | | SPC (rural) | | BO (urban) | | SPC (rural) | |
| | D | N | D | N | D | N | D | N | D | N | D | N |
| **Sulfate** | 2.1 | 2.6 | 1.7 | 2.3 | 0.27 | 0.32 | 0.16 | 0.28 | 2.3 | 2.8 | 1.8 | 2.6 |
| **Nitrate** | 0.24 | 0.47 | 0.32 | 2.40 | 1.6 | 1.7 | 0.85 | 1.5 | 1.8 | 2.1 | 1.2 | 3.9 |
| **Chloride** | 0.023 | 0.043 | 0.005 | 0.041 | 0.17 | 0.27 | 0.05 | 0.14 | 0.13 | 0.27 | 0.06 | 0.19 |
| **Ammonium** | 0.83 | 1.1 | 0.70 | 1.5 | 0.05 | 0.07 | 0.02 | 0.15 | 0.88 | 1.1 | 0.69 | 1.6 |
| **Magnesium** | 0.01 | 0.01 | 0.01 | 0.01 | 0.06 | 0.07 | 0.04 | 0.04 | 0.07 | 0.08 | 0.04 | 0.05 |
| **Calcium** | 0.07 | 0.07 | 0.06 | 0.07 | 0.50 | 0.48 | 0.35 | 0.41 | 0.56 | 0.53 | 0.40 | 0.47 |
| **Sodium** | 0.04 | 0.05 | 0.03 | 0.02 | 0.37 | 0.48 | 0.20 | 0.22 | 0.41 | 0.53 | 0.23 | 0.24 |
| **WSOC** | 1.4 | 1.9 | 0.94 | 1.6 | 0.54 | 0.53 | 0.28 | 0.44 | 1.9 | 2.4 | 1.2 | 2.1 |
| **TC** | 2.5 | 3.5 | 1.3 | 2.2 | 1.5 | 1.5 | 0.8 | 1.4 | 4.1 | 5.2 | 2.1 | 3.5 |



1 **Table 2 – Factor loadings of PCA after Varimax rotation. Only absolute values larger than 0.2 are shown.**
2 **Absolute values larger than 0.60 are considered significant and printed in bold. The number beside each**
3 **chemical species in the first column indicates the impactor stage.**

| Site | SPC | | | | | | BO | | | | | |
|---|---|---|---|---|---|---|---|---|---|---|---|---|
| Rotated component | RC5 | RC1 | RC6 | RC2 | RC4 | RC3 | RC1 | RC2 | RC3 | RC6 | RC4 | RC5 |
| Assigned source | condensation | aqSIA+SOA | dust | Sea salt | RTI var. | gasSIA+gasSOA | seas alt | aqSIA+SOA | gasSIA+gasSOA | resusp. | RTI var. | condensation |
| Nitrate_1 | **0.81** | | | -0.24 | 0.24 | | | | | | | **0.69** |
| Sulfate_1 | -0.32 | 0.28 | | | | **0.79** | | 0.26 | **0.84** | | | |
| Oxalate_1 | -0.34 | | 0.25 | -0.24 | 0.45 | 0.37 | -0.53 | 0.24 | 0.42 | | | |
| Ammonium_1 | | 0.21 | | | 0.23 | **0.84** | | | **0.79** | | 0.29 | |
| WSOC_1 | | 0.23 | 0.35 | -0.4 | 0.29 | 0.46 | -0.45 | 0.37 | 0.41 | 0.3 | 0.38 | |
| Nitrate_2 | **0.79** | 0.36 | | | | | | 0.28 | | | | **0.84** |
| Sulfate_2 | | **0.83** | | 0.3 | | | 0.42 | **0.75** | 0.22 | | | |
| Oxalate_2 | 0.28 | **0.81** | 0.31 | -0.22 | | | | **0.71** | 0.43 | | | 0.34 |
| Sodium_2 | | | | 0.25 | -0.23 | 0.33 | 0.43 | -0.21 | 0.56 | 0.23 | -0.29 | |
| Ammonium_2 | 0.56 | **0.69** | | | | 0.24 | 0.39 | **0.77** | | | | 0.38 |
| WSOC_2 | 0.3 | **0.71** | 0.46 | | | | | **0.72** | 0.43 | | | 0.3 |
| Nitrate_3 | **0.82** | 0.4 | | | | | 0.21 | 0.36 | -0.25 | | | **0.78** |
| Sulfate_3 | 0.36 | **0.79** | | | | | 0.32 | **0.86** | | | | |
| Oxalate_3 | 0.45 | **0.73** | 0.32 | | | | | **0.76** | | | | |
| Sodium_3 | -0.27 | | 0.21 | **0.84** | | | **0.87** | | | | | |
| Ammonium_3 | **0.67** | **0.67** | | | | | 0.29 | **0.81** | | | | 0.32 |
| Magnesium_3 | -0.24 | 0.22 | **0.8** | 0.33 | | | **0.62** | | 0.24 | 0.42 | | 0.28 |
| Calcium_3 | | | **0.92** | | | | | | | 0.5 | -0.37 | 0.27 |
| WSOC_3 | 0.5 | **0.75** | 0.32 | | | | | **0.86** | | | | 0.23 |
| Chloride_4 | 0.57 | | | 0.32 | | -0.29 | 0.51 | 0.22 | -0.45 | | | 0.31 |
| Nitrate_4 | **0.82** | | | 0.43 | | | **0.79** | 0.28 | | 0.28 | | 0.33 |
| Sulfate_4 | **0.69** | 0.32 | | 0.39 | -0.23 | | **0.75** | 0.45 | | 0.24 | -0.3 | |
| Oxalate_4 | | 0.5 | 0.35 | 0.38 | -0.23 | | 0.31 | 0.24 | 0.24 | 0.36 | -0.37 | |
| Sodium_4 | | | | **0.94** | | | **0.9** | | -0.21 | | | |
| Ammonium_4 | **0.89** | 0.27 | | | | | 0.34 | 0.3 | | 0.24 | | **0.7** |
| Magnesium_4 | | | 0.57 | **0.77** | | | **0.88** | 0.21 | | 0.29 | | |
| Calcium_4 | | | **0.93** | | | | 0.41 | | | **0.64** | -0.28 | 0.29 |
| WSOC_4 | 0.47 | 0.31 | 0.52 | | | | | **0.67** | | 0.42 | | -0.28 |
| Chloride_5 | 0.57 | 0.29 | | 0.23 | | | 0.47 | | **-0.66** | 0.26 | | |
| Nitrate_5 | 0.56 | 0.37 | 0.37 | 0.46 | | | 0.54 | | | **0.65** | | |
| Sulfate_5 | 0.47 | | 0.33 | **0.64** | -0.3 | | **0.69** | | | **0.6** | -0.22 | |
| Oxalate_5 | | 0.29 | **0.64** | 0.21 | | | | 0.29 | | 0.55 | -0.46 | |
| Sodium_5 | | -0.23 | | **0.75** | | | **0.71** | | -0.5 | 0.32 | | |
| Ammonium_5 | **0.8** | 0.29 | | | | 0.26 | | 0.23 | | 0.54 | | 0.5 |
| Magnesium_5 | 0.29 | | **0.6** | **0.66** | | | **0.69** | | -0.31 | 0.56 | | |
| Calcium_5 | | | **0.87** | | | 0.24 | 0.33 | | | **0.89** | | |
| WSOC_5 | 0.32 | 0.3 | **0.65** | 0.21 | | | | 0.48 | | 0.55 | | -0.27 |
| RT_waterandice | | | | | **-0.95** | | | | | | **-0.93** | |
| RT_naturalveg | | | | | **0.7** | -0.47 | | | | | **0.82** | -0.21 |
| RT_agriculture | | | | | **0.81** | 0.39 | -0.33 | | | | **0.84** | |
| RT_urbanareas | | | | | **0.9** | | | | | | **0.91** | |
| RT_bareareas | 0.43 | | | | | **0.64** | -0.53 | | | 0.29 | | 0.25 |
| Sunflux_alongtraj | | **0.68** | | | | 0.5 | -0.22 | 0.24 | **0.68** | | -0.25 | |
| Temperature | **-0.85** | | | | | 0.3 | | -0.29 | **0.69** | | -0.26 | -0.23 |
| RH | **0.82** | | | | | -0.31 | | **0.6** | -0.49 | | | 0.34 |
| Explained variance (%) | 22 | 15 | 14 | 12 | 8 | 8 | 19 | 17 | 11 | 11 | 10 | 9 |
| Cumulative variance (%) | 22 | 37 | 51 | 63 | 71 | 79 | 19 | 36 | 47 | 58 | 68 | 77 |



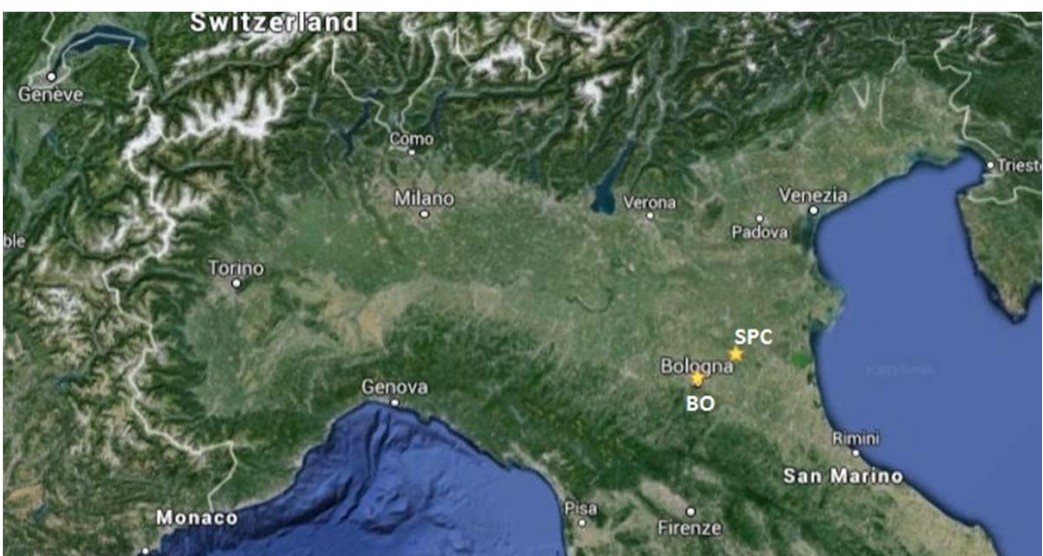

2  **Fig. 1 – Location of the sampling sites in the Po Valley.**





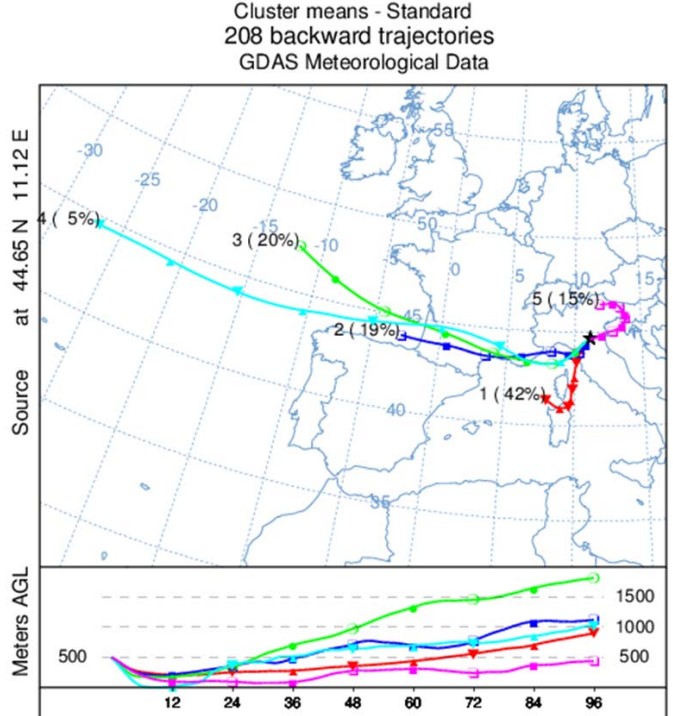

2    **Fig. 2 - Map with average trajectories for each obtained cluster with, in brackets, corresponding percentage of**
3    **occurrence. The numbers outside the brackets identify each cluster. Figure refers to 96h trajectories arriving at**
4    **500 m a.g.l.**





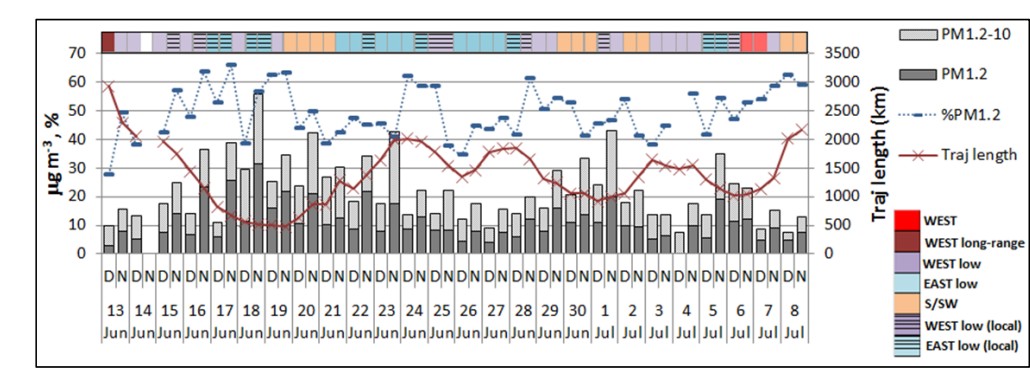

**Fig. 3 - Time series of PM$_{1.2}$ and PM$_{10}$ mass concentrations (in µg m$^{-3}$) and of the PM$_{1.2}$ to PM$_{10}$ ratio (%) for**
**SPC. 4-days (96h) back trajectory length (km) is plotted superimposed to the graph, while air mass**
**classification (in colors) is reported on top of it. The samples are labelled according to collection starting**
**date, with "D" and "N" denoting respectively daytime and night-time samples.**



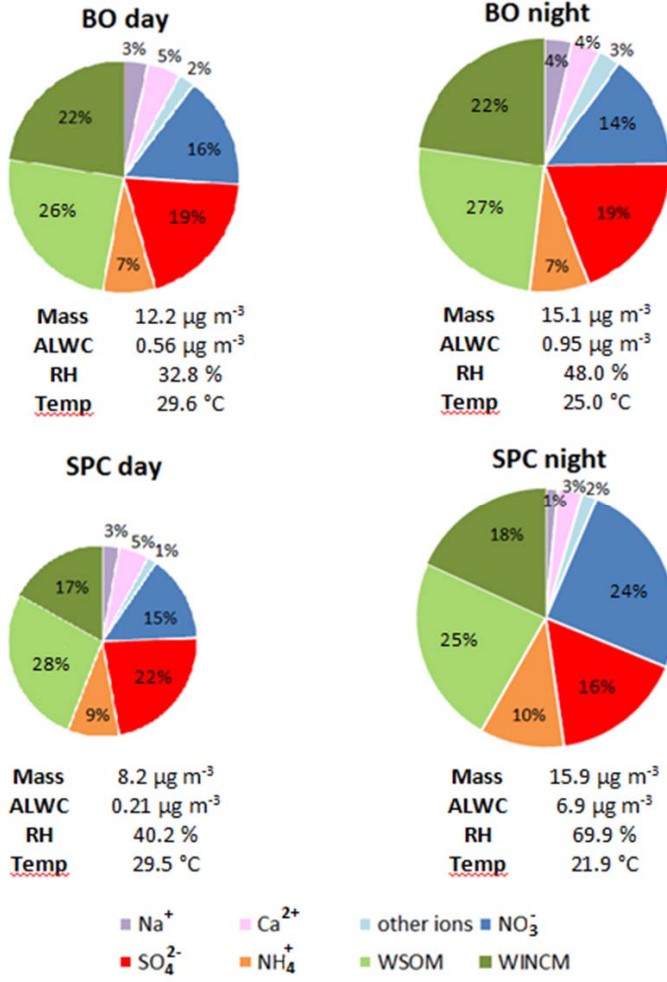

**Fig. 4 – Average day and night PM$_{10}$ composition at BO and SPC during the campaign. "Other ions" include chloride, nitrite, potassium and magnesium. WSOM stands for water soluble organic matter, while WINCM for water insoluble carbonaceous matter. Average mass, ALWC, RH and temperature are indicated below each pie. The size of each pie chart is proportional to the total measured mass reported.**





2    **Figure 5 – Time series of sulfate, nitrate and WSOC size-segregated concentrations in BO and SPC. Please note**
3    **the different scale for nitrate and WSOC in BO.**





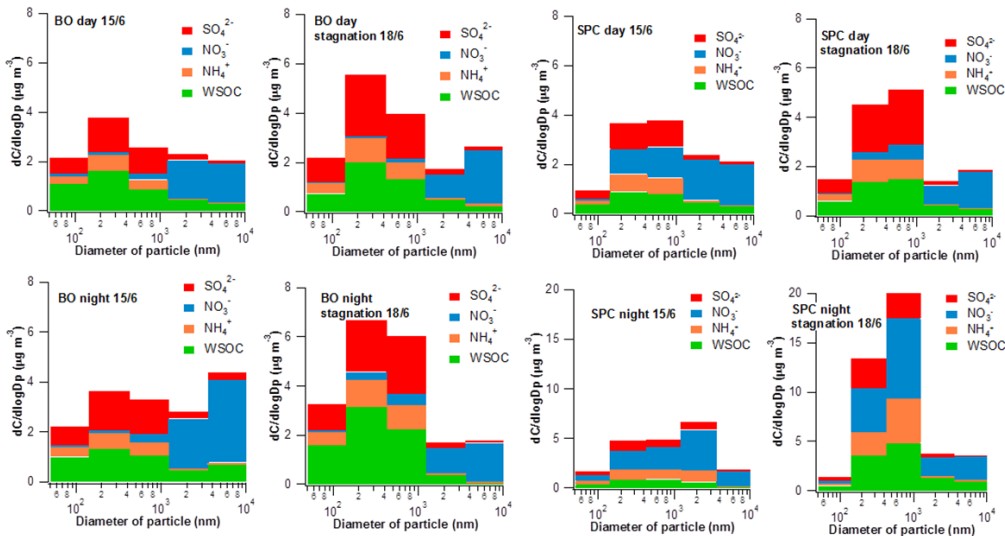

2 **Fig. 6 - Size-resolved aerosol composition for BO and SPC during day (top) and night (bottom) respectively**
3 **during one day characterized by background conditions (15/6) and during one day under stagnant conditions**
4 **(18/6) (notice the different scale for concentrations in the bottom right panel).**





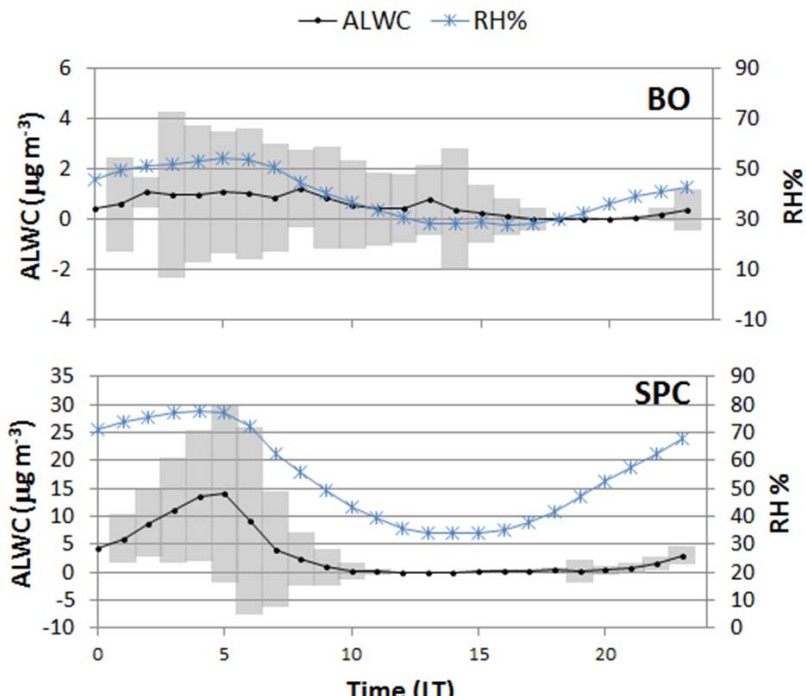

2 **Fig. 7 – Average diurnal variation of Relative Humidity (RH) and Aerosol Liquid Water Content (ALWC)**
3 **during the campaign. Please note the different scale of ALWC at the two sites. The shaded area denotes ± 1**
4 **standard deviation for ALWC concentrations.**





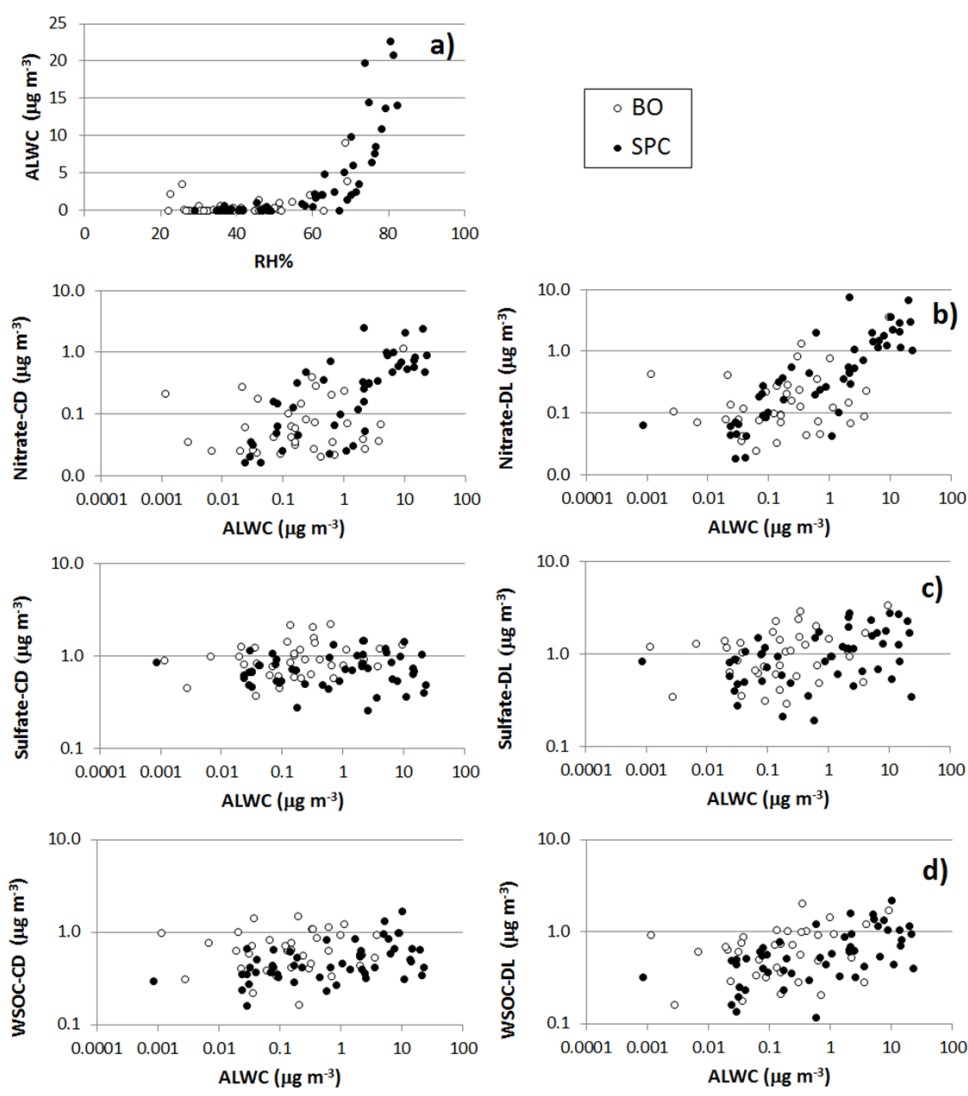

2    **Fig. 8 – Relationship between ALWC and RH% (a), and between nitrate (b), sulfate (c) and WSOC (d) in the**
3    **condensation (CD, left panel) and in the droplet (DL, right panel) mode of particles, and ALWC averaged over**
4    **the sampling periods of the Berner Impactors.**





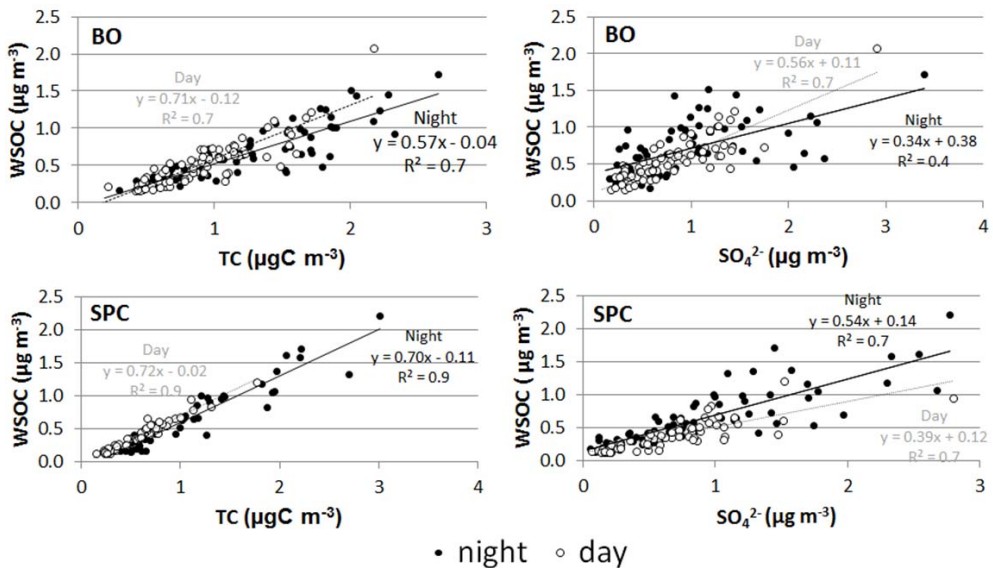

2    **Fig. 9 - Linear regressions between WSOC and TC (left) and WSOC and SO$_4^{2-}$ (right) in the size intervals: 1)**
3    **0.05 – 0.14 µm, 2) 0.14 – 0.42 µm, and 3) 0.42 – 1.2 µm in daytime and at night for BO (top) and SPC (bottom).**





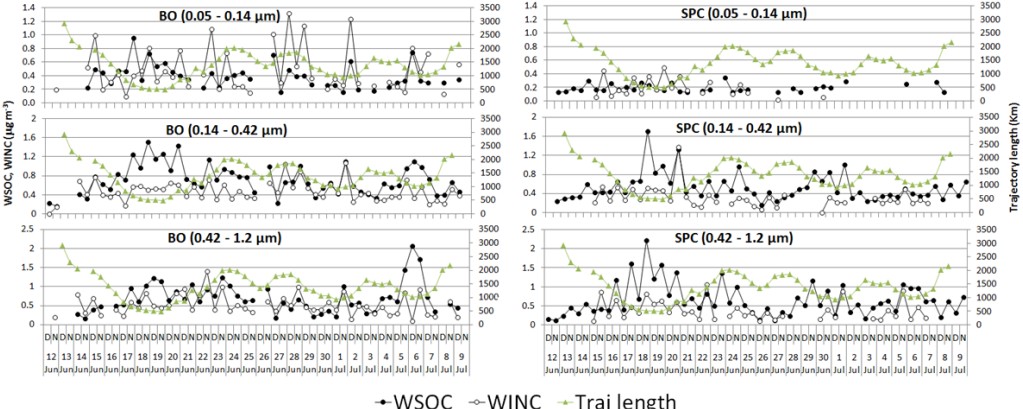

3    **Fig. 10 – Time-series of WSOC and WINC (=TC-WSOC) concentrations in the size intervals: 1) 0.05 – 0.14 μm,**
4    **2) 0.14 – 0.42 μm, and 3) 0.42 – 1.2 μm. 4 days back-trajectories length is superimposed to each graph. Results**
5    **for the urban (BO) and for the rural (SPC) station are shown.**





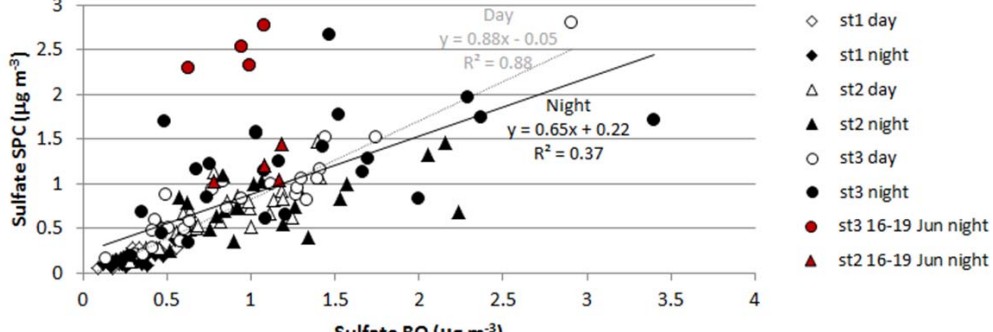

**Fig. 11 – Scatterplot of sulfate concentration at SPC vs BO during day and night for the impactor stages 1 (0.05 –**
**0.14 μm), 2 (0.14 – 0.42 μm) and 3 (0.42 – 1.2 μm). The regression lines are referred to the diurnal (gray line) and**
**nocturnal (black line) concentrations in the three stages as a whole. Condensation and droplet mode samples for**
**the stagnant nights 16-19 June are filled in red.**

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
