# Peer review of "Size-resolved aerosol composition at an urban and a rural site in"

_Atmospheric Chemistry and Physics, 2015_

## Referee Comment (RC1) · Anonymous Referee #1 · 13 Apr 2016

General comments

The manuscript presents size-segregated aerosol measurements made with a Berner Impactor in summer 2012 at a rural and an urban site each in the Po Valley, Italy. Sampled particles were chemically analyzed for inorganic and organic species, and Principal Component Analysis was done for the differentiation of aerosol components. This is a well written paper that presents a careful analysis of an interesting dataset. I thus recommend publication after the comments below have been addressed. Quite a few comments are formulated as questions, which are meant as encouragement for the authors to add the answer to the manuscript in the form of additional explanations.

Specific comments

Abstract: Mention which method you used to do the chemical analysis in the abstract.

P. 2, l. 20: Specify quasi-ultrafine

P. 3, l. 1-2: This statement is not entirely correct. The sampling efficiency can be close to unity for particles with a diameter up to 2.5 $\mu$m when using a high-pressure aerodynamic lens (http://www.atmos-meas-tech.net/6/3271/2013/amt-6-3271-2013-discussion.html).

P. 3, l. 24: Say in a sentence or two what the Lenschow perspective is, so that the reader gets a basic idea without having to consult the reference.

P. 3, l. 37 – p. 4, l. 6: How is the predominating wind direction in the Po Valley (and how was it during the measurement campaign? Was SPC mainly downwind of BO, or were both sites downwind of Milano? Were the air masses ever coming from the East (sea)? Please add a few sentences here.

P. 4, l. 18 – 30: Is there any information on the sampling efficiency (e. g. bouncing of particles from the impactor)? And is there a large influence from gaseous organic compounds being deposited on the aluminum foils? I imagine the mass of semi- or low volatile gaseous material deposited during a 12 hr sampling period could add substantial signal to the $CO_2$ signal. Please elaborate.

P. 6, l. 22 – 27: During the periods with the two most prevalent clusters (1 and 5) – were the meteorological/ boundary layer conditions such that the measurement stations were heavily affected by these air masses? Inversion layers and/or low boundary layer conditions would make very local air masses more important. Please elaborate and add data if possible.

P. 7, l. 1 – 2: High PM mass loadings go together with shorter trajectory lengths, a more regional/local aerosol, and a higher fraction of the smaller particles. Based on this shift in size distribution you make the assumption that there is a larger contribution from secondary components. Couldn't there also be a larger contribution from

primary emissions (e. g. soot or primary organic particles from traffic or e. g. biomass burning)?

P. 7, l 19 − 22: The partitioning of ammonium nitrate (AN) into the gas phase shows a strong temperature dependence (as you mention a few times in the manuscript). If the T difference between day and nights were significant, this could be a reason for the increased AN contributions at night. Mention this already here.

P. 7, l. 34 − 35: Ammonia concentrations could be higher at the rural site due to agricultural emissions, leading to higher AN formation. You mention this on p. 10, l. 35 − 37. I would add a similar statement here.

P. 8, l. 33 − p. 9, l. 4: Can you say anything about boundary layer heights for the two stagnant periods? It is rather surprising to me that the shorter period, when air masses would have less time to accumulate, showed higher concentrations. How do the ALWC for the 2 periods compare? RH conditions seem to have been different for the two periods (also compare paragraph p. 10, l. 28 − 39).

P.9, l. 23 − 25: This statement here is very vague and without much empirical evidence. I suggest removing it from here.

P. 10, l. 5: The numbers are reported for 25°C. What were the night time temperatures, and the respective RHD?

P. 10, l. 18 -21: The authors show in Figure 8 the correlation of e. g. nitrate and ALWC, however discuss it as "relationship between nitrate or sulfate and RH". As a correlation of especially nitrate and RH is much less meaningful (both are inversely related to temperature and thus higher at night, which probably drives the correlation) than nitrate and ALWC, to which the authors seem to agree, at least according to Fig. 8, I suggest revising that paragraph accordingly.

P. 11, l. 25 − 30: With WSOA also of predominantly secondary origin, the common regional nature of WSOC and SO4 can be a reason for their correlated temporal patterns.

P. 13, l. 12: There is quite a large fraction that remains unexplained with the choice of 6 principal components. Please add a short paragraph (here or in the supplementary section) on how this unexplained fraction depends on the number of principal components.

P. 13, l. 26 − 31: RC1 is not only related to photochemical production, but is also of regional nature (see previous comment). This is an important statement to make as it has implications not only for science, but also for air quality policy makers – regional aerosol is much more difficult to control due to the large area of formation. This is only mentioned in "Discussions and conclusions", but I suggest adding it here already.

P. 14, l. 1 − 6: Is there a correlation of this component with wind directions from the sea? If you have such data it would make a stronger statement for the identification of this component.

Technical comments

P. 2, l. 19 : WINC?

P. 3, l. 27: "such as" gives the impression of the authors having tried several approaches, which, however, are not mentioned. Please revise.

P 4., l. 21: TOC-VCPH?

P. 8, l. 11: "Figs" instead of "Figg"

P. 14, l. 19: Should be "at minimum"

---

## Referee Comment (RC2) · Anonymous Referee #2 · 3 May 2016

This manuscript represents a thorough analysis of size-resolved aerosol composition measurements from two sites in the Po Valley during a summer campaign. In the introduction, the authors mention that the two site approach was intended to be used so that the rural site could serve as the 'background' for the urban measurements. In fact, the authors demonstrate that the higher relative humidity during the night at the rural site leads to substantially higher ammonium nitrate (and WSOC) at this rural site. In general, I thought the paper was clearly written, but it would be useful for the others to revisit this 'Lenschow perspective' in the conclusions and comment on whether a rural site can really be interpreted as the background for a nearby urban site. I also include some minor questions and suggestions:

Page 5, Line 9 – 'which did not justify the presence of ALW on particles' should be reworded to 'which prevented accurate calculation of ALW on particles'

Section 2.3 – Can the authors confirm whether inorganic carbonate salts have the ability to contribute to the WSOC reported by their measurement protocol? If not, in which category would the mineral dust carbonate be counted?

Section 2.4 – Which site was used as the origin of the back trajectories?

Page 11, Lines 25-30 – The moderately good correlation coefficient of 0.7 between WSOC and sulfate could be due to shared photochemical sources, but it may have other origins as well. Given that the values for each constituent range from 0.2 – 2, and there probably isn't a factor of ten variability between photon flux between days, there's likely at least one other factor driving the shared variability.

Section 3.5 – The numbering of the rotated components doesn't make sense to the reader. Also, the last three components that are discussed are not identified by the numbering system used in Table 2.

Page 14, Lines 35-38 – Could the higher nighttime droplet mode sulfate measurements at SPC be the results of more cloud processing of SO2 as a result of higher RH?

I suggest adding a scale bar to Figure 1.

Figure 7 – it doesn't really make sense to have negative ALWC values. I would recommend using the boxes to show the interquartile range, or the 10th and 90 percentiles, rather than +/- the standard deviation.

Figure 8, panel a – it would be useful to have the y-axis of this figure on a log scale as well

[Figure]

---

## Author Comment (AC1) · 9 Jun 2016

Reviewer 1

**General comments**

The manuscript presents size-segregated aerosol measurements made with a Berner Impactor in summer 2012 at a rural and an urban site each in the Po Valley, Italy. Sampled particles were chemically analyzed for inorganic and organic species, and Principal Component Analysis was done for the differentiation of aerosol components. This is a well written paper that presents a careful analysis of an interesting dataset.

I thus recommend publication after the comments below have been addressed. Quite a few comments are formulated as questions, which are meant as encouragement for the authors to add the answer to the manuscript in the form of additional explanations.

**Specific comments**

Abstract: Mention which method you used to do the chemical analysis in the abstract.

This information has been added in the abstract in lines 1-4 which changed into:

"The aerosol size-segregated chemical composition was analyzed at an urban (Bologna) and a rural site (San Pietro Capofiume) in the Po Valley, Italy, during June and July 2012, by ion-chromatography (major water soluble ions and organic acids) and evolved gas analysis (total and water soluble carbon), to investigate sources and mechanisms of secondary aerosol formation during the summer".

P. 2, l. 20: Specify quasi-ultrafine

Done

P. 3, l. 1-2: This statement is not entirely correct. The sampling efficiency can be close to unity for particles with a diameter up to 2.5 μm when using a high-pressure aerodynamic lens (http://www.atmos-meas-tech.net/6/3271/2013/amt-6-3271-2013-discussion.html).

Thank you for noticing this, indeed the sentence was not well formulated. Lines 1-2 have been changed into

"However, aerosol mass spectrometers (AMS) generally suffer of poor sensitivity for thermally refractory compounds and could not be deployed for the analysis of coarse particles (>2.5 μm) chemical composition."

P. 3, l. 24: Say in a sentence or two what the Lenschow perspective is, so that the reader gets a basic idea without having to consult the reference

Lines 24-26 have been changed into:

"The two-site approach was adopted to estimate the contribution and composition of rural background particles with respect to the urban contribution, according to the Lenschow perspective (Lenschow et al., 2001), based on the assumption that PM concentration measured at an urban location is the result of the addition of regional background, urban contribution given by the sources inside the agglomeration, and road traffic for near streets sites".

**P. 3, l. 37 – p. 4, l. 6: How is the predominating wind direction in the Po Valley (and how was it during the measurement campaign? Was SPC mainly downwind of BO, or were both sites downwind of Milano? Were the air masses ever coming from the East (sea)? Please add a few sentences here.**

The west-east orientation of the valley favours westerly or easterly circulations, hence from either the inner Po Valley (including Lombardy and Milan) or the Adriatic Sea. During the present study, weak westerly breezes affected the sampling sites at night and in morning hours, while short, intense easterly breezes flowed during the late afternoon (Fig. 2 in Wolf et al., 2015). However, in days of stronger synoptic forcing, a wind pattern characterized by strong easterly winds (persisting all the day) was observed (especially in the second half of June), while a wind pattern characterized by SW winds from the Apennines was also common in June and accounted for most of the days in July until the end of the campaign. In the presence of SW winds, SPC can result downwind of BO and an influence from the urban area cannot be excluded. Under the same conditions, a possible inflow of marine air from the Ligurian Sea is also possible. We will add references to the Wolf et al., study for the breeze circulation, and add up information of wind regimes (and references to a paper in preparation).

This additional information has been added partly in section 2.1 "Sampling sites" and partly in section 3.1 "Back-trajectory patterns".

**P. 4, l. 18 – 30: Is there any information on the sampling efficiency (e. g. bouncing of particles from the impactor)? And is there a large influence from gaseous organic compounds being deposited on the aluminium foils? I imagine the mass of semi- or low volatile gaseous material deposited during a 12 hr sampling period could add substantial signal to the CO2 signal. Please elaborate**

A low pressure impactor such as the Berner Impactor can suffer for sampling artifacts, which are mostly represented by loss of particles due to bouncing, or evaporation of semivolatile compounds during periods characterized by elevated temperatures. Positive artifacts due to adsorption of gaseous organic species represent a problem when employing filters where the air flow is forced through a fibrous material with a high surface area, but they are of limited importance in impactors where aerosol particles are collected by their inertia, while gaseous molecules follow the air flow: in principle, they should diffuse against a pressure gradient to reach the sampling foil.

In order to evaluate potential negative artifacts, the size-integrated ($PM_{1.2}$, sum of the first three stages) impactor concentrations of sulfate, nitrate and ammonium were compared to those obtained by another co-located off-line system, a High Volume Digitel PM1 sampler, and by HR-ToF-AMS (PM<1) measurements, as reported in the supplementary material (Figs. S7 and S8). The HiVol system is not affected by bouncing but can be affected by semivolatile losses, while the AMS is believed not affected by the loss of semivolatile species (with a vapor pressure equal or lower than that of ammonium nitrate).

To quantify the particle losses due to bouncing it is useful to compare sulfate concentrations, which is not volatile, for diurnal samples. Bouncing in fact is more effective at low relative humidity. The simple linear regression of Berner Impactor concentrations versus the two co-located instruments, when forced to the origin, highlights a 15% loss with respect to both High Volume ($R^2 = 0.73$) and AMS ($R^2 = 0.75$), which could be attributed to bouncing. More precisely, since High Volume quartz fibre filters can also absorb $SO_2$, this 15% would represent the extent of negative artifacts on impactors due to bouncing, assuming that no positive artifacts from $SO_2$ absorption occurred on the HiVol.

**P. 6, l. 22 – 27: During the periods with the two most prevalent clusters (1 and 5) – were the meteorological/ boundary layer conditions such that the measurement stations were heavily affected by these air masses? Inversion layers and/or low boundary layer conditions would make very local air masses more important. Please elaborate and add data if possible.**

We have revised Fig. 2, which did not reflect the clustering of 1456 trajectories. The most represented clusters are clusters 1 and 3, corresponding to short trajectories and low wind speed. It is true that even more local circulations are expected when low-altitude stratifications occur, which is normal for night-time hours in this region. However, in daytime, the PBL extends easily up to 1500 or 2000 m above the ground level (see Figure below derived from Lidar measurements at SPC), allowing the regional scale circulation (well traced by back-trajectories) impact the site.

**P. 7, l. 1 – 2: High PM mass loadings go together with shorter trajectory lengths, a more regional/local aerosol, and a higher fraction of the smaller particles. Based on this shift in size distribution you make the assumption that there is a larger contribution from secondary components. Couldn't there also be a larger contribution from primary emissions (e. g. soot or primary organic particles from traffic or e. g. biomass burning)?**

The statement in P.7 lines 1-2 has been deleted, since the issue of secondary aerosol formation is addressed later on in the manuscript through the analysis of chemical composition. The previous sentence (P.6 lines 39-40) has been modified as follows:

"The aerosol mass increased from 15 to 18 June, with an enhanced contribution of the PM1.2 fraction on the total PM10, together with a sharp decrease of the trajectories length, following the onset of an anticyclonic period with low wind and air stagnation over the Po Valley".

**P. 7, l 19 – 22: The partitioning of ammonium nitrate (AN) into the gas phase shows a strong temperature dependence (as you mention a few times in the manuscript). If the T difference between day and nights were significant, this could be a reason for the increased AN contributions at night. Mention this already here.**

The following sentence has been added at pag 7 line 22

"The lower average nocturnal temperatures (22°C versus 25°C), and the higher average nocturnal relative humidity (70% versus 48%) of SPC with respect to BO probably played an important role".

**P. 7, l. 34 – 35: Ammonia concentrations could be higher at the rural site due to agricultural emissions, leading to higher AN formation. You mention this on p. 10, l. 35 – 37. I would add a similar statement here.**

Done. Lines 37-39 have been changed into:

"By contrast at night the aerosol mass was similar at the two sites but the chemical composition was different, with an enrichment of ammonium nitrate at the rural site, probably also favoured by the high ammonia concentrations from agricultural sources, which during the campaign were measured only at the rural site (Sullivan et al., 2015).

**P. 8, l. 33 – p. 9, l. 4: Can you say anything about boundary layer heights for the two stagnant periods? It is rather surprising to me that the shorter period, when air masses would have less time to accumulate,**

**showed higher concentrations. How do the ALWC for the 2 periods compare? RH conditions seem to have been different for the two periods (also compare paragraph p. 10, l. 28 – 39).**

During the two stagnation periods, weak winds were accompanied by a shallower Boundary Layer, as observed in the following plot reporting PBL height from Lidar measurements and from radiosoundings in SPC (Bucci et al., in preparation). PBL was around 1000 m or lower at noon under stagnant conditions, compared to 1600-1700 m observed during other periods of the campaign at the same hour of the day.

[Figure]

The ALWC in SPC was not very different between the two stagnation periods. In both cases high nocturnal levels of liquid water were observed (about 50 µg m$^{-3}$ during the first event and 60 µg m$^{-3}$ during the second). In contrast, liquid water was almost absent in BO during the first event while it reached a maximum (17 µg m$^{-3}$) during the second event in correspondence to the highest diurnal and nocturnal relative humidity (cfr. Fig. S7, which has been added in the Supplementary Material). Indeed the two days 5 and 6 July were characterized by higher relative humidity, especially during the day. Compared to the rest of the campaign which was characterized by clear sky, during these two days in July scattered clouds were observed and some showers occurred in the northern part of Italy, which could have contributed to an enhancement of relative humidity over the whole Po Basin.

**P.9, l. 23 – 25: This statement here is very vague and without much empirical evidence. I suggest removing it from here.**

The sentence has been removed.

**P. 10, l. 5: The numbers are reported for 25°C. What were the night time temperatures, and the respective RHD?**

During the study period the hourly nocturnal temperatures and corresponding DRH for ammonium nitrate and ammonium sulfate (in brackets) ranged from 12°C (69.3% for ammonium nitrate and 80.7% for ammonium sulfate) to 30°C (58.6% for ammonium nitrate and 79.5 for ammonium sulfate) in SPC and from 16°C (67.1% for ammonium nitrate and 80.3 for ammonium sulfate) to 30°C in BO. The text has been modified including this additional information on the temperature dependence of DRH.

**P. 10, l. 18 -21: The authors show in Figure 8 the correlation of e. g. nitrate and ALWC, however discuss it as "relationship between nitrate or sulfate and RH". As a correlation of especially nitrate and RH is much less meaningful (both are inversely related to temperature and thus higher at night, which probably drives the correlation) than nitrate and ALWC, to which the authors seem to agree, at least according to Fig. 8, I suggest revising that paragraph accordingly.**

Our mistake. A preliminary version of Figure 8 reported the correlations with RH% instead of ALWC. We have corrected the text. Thank you for noticing it.

**P. 11, l. 25 – 30: With WSOA also of predominantly secondary origin, the common regional nature of WSOC and SO4 can be a reason for their correlated temporal patterns.**

We agree, and the common regional source is a good explanation for the good correlation between WSOC and sulfate at both BO and SPC during daytime, when photochemistry is expected to be a main source for these two types of compounds. Conversely, different degrees of association between WSOC and sulfate are observed at the two sites at night-time: the correlation is lost in BO, as expected in the absence of photochemical production, but it is still good in SPC, where another nocturnal common source must be hypothesised.

**P. 13, l. 12: There is quite a large fraction that remains unexplained with the choice of 6 principal components. Please add a short paragraph (here or in the supplementary section) on how this unexplained fraction depends on the number of principal components.**

Increasing the number of factors progressively improves the explained variance, but each additional factor contributes only a small fractional increase: with a number of factors to 7, 8, or 9, the total explained variance would increase to only 82, 85, and 87%, respectively. Interpretation of the additional factors is also challenging. For the above reasons, a 6-factor solution was chosen as the best one.

The text has been modified accordingly including this additional information.

**P. 13, l. 26 – 31: RC1 is not only related to photochemical production, but is also of regional nature (see previous comment). This is an important statement to make as it has implications not only for science, but also for air quality policy makers – regional aerosol is much more difficult to control due to the large area of formation. This is only mentioned in "Discussions and conclusions", but I suggest adding it here already.**

The following sentence has been added after line 31:

"The significant contribution represented by this source on the total variance described for the two datasets highlights the importance of regional-scale secondary aerosol formation processes for the Po Valley environment"

.. and at the end of the Discussions and Conclusions we will append:

"In conclusion, the characteristics of the size-segregated aerosol compositions and its variability at a rural site and at a urban background site in the Po Valley could be explained by a limited number of factors reflecting main physico-chemical processes and/or transport patterns in the atmosphere. For accumulation mode particles in particular, our analysis points to two main processes: "(1) The *photochemical SIA and SOA* which occur at comparable concentrations between BO and SPC and are particularly evident in daytime hours when the lower atmosphere is well mixed, and indicating that a major fraction of background submicron aerosol concentrations in the Po Valley actually originates from regional-scale sources, which can extend over vast continental areas (see also Fig. S12 in Decesari et al., Atmos. Chem. Phys., 14, 12109-12132, 2014). This has implications for air quality mitigation, because this photochemical component is expected to show little sensitivity to local-scale (city-level) regulations. (2) *Nocturnal SIA and SOA* whose formation is enhanced in the shallow, cool and humid boundary layer and mediated by the presence of aerosol liquid water. Such component of the rural background aerosol appears more volatile (hence labile) and more heterogeneously distributed across the Po Valley, with the inner part (where most agricultural activity reside) acting as a source region respect to its southern periphery (more urbanized). The rural

background concentration level is therefore variable, somewhat "tilted" horizontally across the Valley, at least for half of the day. These results represent an example of a limitation to the classical Lenschow model."

**P. 14, l. 1 – 6: Is there a correlation of this component with wind directions from the sea? If you have such data it would make a stronger statement for the identification of this component.**

The following plot shows the scores of this factor (formerly RC2, but now the principal components have been renumbered and this component corresponds to RC4 in SPC and RC1 in BO) for BO and SPC. The highest scores of this component occur on June 21 and on 2-3 July. An observation of the air mass classification reported on top of Fig. 3 indicates for those days an air mass origin from the Tyrrenian Sea (South/South-West), which extended down to the North African continent in correspondence of the maximum scores of this component. This principal component would therefore consist of a mixture of sea salt contribution and desert dust, which is reported as enriched in K+, Mg2+, Ca2+ and Na+. This is in agreement with lidar observations of Saharan Dust events during the campaign, described in Bucci et al., (in preparation).

---

## Author Comment (AC2)

**Reviewer 2**

This manuscript represents a thorough analysis of size-resolved aerosol composition measurements from two sites in the Po Valley during a summer campaign. In the introduction, the authors mention that the two site approach was intended to be used so that the rural site could serve as the 'background' for the urban measurements. In fact, the authors demonstrate that the higher relative humidity during the night at the rural site leads to substantially higher ammonium nitrate (and WSOC) at this rural site. In general, I thought the paper was clearly written, but it would be useful for the others to revisit this 'Lenschow perspective' in the conclusions and comment on whether a rural site can really be interpreted as the background for a nearby urban site.

We have added this paragraph to the Discussion and Conclusions to clarify the implications of this two-site experiment:

"In conclusion, the characteristics of the size-segregated aerosol compositions and its variability at a rural site and at a urban background site in the Po Valley could be explained by a limited number of factors reflecting main physico-chemical processes and/or transport patterns in the atmosphere. For accumulation mode particles in particular, our analysis points to two main processes: "(1) The *photochemical SIA and SOA* which occur at comparable concentrations between BO and SPC and are particularly evident in daytime hours when the lower atmosphere is well mixed, and indicating that a major fraction of background submicron aerosol concentrations in the Po Valley actually originates from regional-scale sources, which can extend over vast continental areas (see also Fig. S12 in Decesari et al., Atmos. Chem. Phys., 14, 12109-12132, 2014). This has implications for air quality mitigation, because this photochemical component is expected to show little sensitivity to local-scale (city-level) regulations. (2) *Nocturnal SIA and SOA* whose formation is enhanced in the shallow, cool and humid boundary layer and mediated by the presence of aerosol liquid water. Such component of the rural background aerosol appears more volatile (hence labile) and more heterogeneously distributed across the Po Valley, with the inner part (where most agricultural activity reside) acting as a source region respect to its southern periphery (more urbanized). The rural background concentration level is therefore variable, somewhat "tilted" horizontally across the Valley, at least for half of the day. These results represent an example of a limitation to the classical Lenschow model."

**I also include some minor questions and suggestions:**

**Page 5, Line 9 – 'which did not justify the presence of ALW on particles' should be reworded to 'which prevented accurate calculation of ALW on particles'**

Done

**Section 2.3 – Can the authors confirm whether inorganic carbonate salts have the ability to contribute to the WSOC reported by their measurement protocol? If not, in which category would the mineral dust carbonate be counted?**

When measuring TC on liquid samples (water aerosol extracts in this case) the instrument performs two distinct analyses on each sample: total carbon (TC) is measured by combustion at 680°C in the presence of a catalyst to become $CO_2$, which is measured by a non-dispersive infrared gas analyser (NDIR); then inorganic carbon (IC) is measured by introducing another aliquot in a reaction vessel where, after acidification, IC is decomposed to $CO_2$, similarly quantified by NDIR. The organic fraction (WSOC) is obtained

by subtracting IC from TC. it is worth specifying that in the water soluble fraction of aerosol the contribution of inorganic carbon was most of times under detection limit for Po Valley aerosol samples.

This information has been added in Section 2.3.

**Section 2.4 – Which site was used as the origin of the back trajectories?**

The used trajectories have been calculated for SPC. Based on the horizontal resolution of the GDAS meteorological data used in the calculation by the HYSPLIT model, i.e. 1°, which corresponds to ~ 100 km × 100 km. The SPC trajectories can be considered valid for BO most of the time (as BO is only 30 km distant from SPC), but SPC, being located on a terrain with a simple orography, is probably an easier "target" for back-trajectory analysis.

**Page 11, Lines 25-30 – The moderately good correlation coefficient of 0.7 between WSOC and sulfate could be due to shared photochemical sources, but it may have other origins as well. Given that the values for each constituent range from 0.2 – 2, and there probably isn't a factor of ten variability between photon flux between days, there's likely at least one other factor driving the shared variability.**

This is probably due to the fact that sulfate and at least a fraction of WSOC are stable photochemical products (at least as WSOC carbon mass, not necessarily molecular composition). Their concentration will therefore depend on a multi-day air mass history and, since back-trajectories analysis shows that air masses were very diverse during the experiment (Fig. 2), this could be the reason for the order of magnitude of variation in concentrations. The regional-scale nature of photochemical SIA and SOA was better clarified in the text.

**Section 3.5 – The numbering of the rotated components doesn't make sense to the reader. Also, the last three components that are discussed are not identified by the numbering system used in Table 2.**

In order to make the PCA results easier to read we changed the numbering of the rotated components for both sites in Tab. 2, just ordering them by decreasing values of explained variance. The numbers on the text have been changed accordingly.

**Page14, Lines35-38–Could the higher nighttime droplet mode sulfate measurements at SPC be the results of more cloud processing of SO2 as a result of higher RH?**

During all the campaign the sky was generally cloud free, with the exception of the period preceding the days 5-6 July, when clouds developed west of the sites and some showers occurred in the Northern part of Italy, though only scattered clouds were present at the two sites. Cloud processing of $SO_2$ can be hypothesized only for this second stagnant period.

**I suggest adding a scale bar to Figure 1.**

Done

**Figure 7 – it doesn't really make sense to have negative ALWC values. I would recommend using the boxes to show the interquartile range, or the 10th and 90 percentiles, rather than +/- the standard deviation.**

The ALWC plots for both sites have been changed into boxplots showing the interquartile range.

**Figure 8, panel a – it would be useful to have the y-axis of this figure on a log scale as well**

Done